# A point mutation decouples the lipid transfer activities of microsomal triglyceride transfer protein

**Meredith H. Wilson**[1], **Sujith Rajan**[2], **Aidan Danoff**[1,3], **Richard J. White**[4,5], **Monica R. Hensley**[1], **Vanessa H. Quinlivan**[1], **Rosario Recacha**[6], **James H. Thierer**[1,3], **Frederick J. Tan**[1], **Elisabeth M. Busch-Nentwich**[4,5], **Lloyd Ruddock**[6], **M. Mahmood Hussain**[2]*, **Steven A. Farber**[1,3]*

1 Department of Embryology, Carnegie Institution for Science, Baltimore, Maryland, United States of America, 2 New York University Long Island School of Medicine, Mineola, New York, United States of America, 3 Department of Biology, Johns Hopkins University, Baltimore, Maryland, United States of America, 4 Wellcome Sanger Institute, Wellcome Genome Campus, Hinxton, Cambridge, United Kingdom, 5 Cambridge Institute of Therapeutic Immunology & Infectious Disease, Department of Medicine, University of Cambridge, Cambridge, United Kingdom, 6 Faculty of Biochemistry and Molecular Medicine, University of Oulu, Oulu, Finland

* Mahmood.Hussain@nyulangone.org (MMH); farber@carnegiescience.edu (SAF)

**Data Availability Statement:** Sequence data were deposited in European Nucleotide Archive under accession ERP023267. Lipidomics data and all

## Abstract

Apolipoprotein B-containing lipoproteins (B-lps) are essential for the transport of hydrophobic dietary and endogenous lipids through the circulation in vertebrates. Zebrafish embryos produce large numbers of B-lps in the yolk syncytial layer (YSL) to move lipids from yolk to growing tissues. Disruptions in B-lp production perturb yolk morphology, readily allowing for visual identification of mutants with altered B-lp metabolism. Here we report the discovery of a missense mutation in microsomal triglyceride transfer protein (Mtp), a protein that is essential for B-lp production. This mutation of a conserved glycine residue to valine (zebrafish G863V, human G865V) reduces B-lp production and results in yolk opacity due to aberrant accumulation of cytoplasmic lipid droplets in the YSL. However, this phenotype is milder than that of the previously reported L475P *stalactite* (*stl*) mutation. MTP transfers lipids, including triglycerides and phospholipids, to apolipoprotein B in the ER for B-lp assembly. *In vitro* lipid transfer assays reveal that while both MTP mutations eliminate triglyceride transfer activity, the G863V mutant protein unexpectedly retains ~80% of phospholipid transfer activity. This residual phospholipid transfer activity of the G863V *mttp* mutant protein is sufficient to support the secretion of small B-lps, which prevents intestinal fat malabsorption and growth defects observed in the *mttp*<sup>stl/stl</sup> mutant zebrafish. Modeling based on the recent crystal structure of the heterodimeric human MTP complex suggests the G865V mutation may block triglyceride entry into the lipid-binding cavity. Together, these data argue that selective inhibition of MTP triglyceride transfer activity may be a feasible therapeutic approach to treat dyslipidemia and provide structural insight for drug design. These data also highlight the power of yolk transport studies to identify proteins critical for B-lp biology.

other source data is provided as Supporting Information files.

**Funding:** This work was supported by National Institutes of Health grants R01 DK093399 (Farber, PI; Busch-Nentwich, Co-PI), R01 GM63904 (The Zebrafish Functional Genomics Consortium; Ekker, PI, Farber, Co-PI), HL-137202 (Hussain, PI), R01 DK121490 (Hussain, PI), R01 HD094778 (Hussain, PI), F31HL139338 to J.H.T and F32DK109592 to M.H.W. (https://www.nih.gov/), as well as G. Harold & Leila Y. Mathers Foundation (Farber, PI) (http://www.mathersfoundation.org/), VA Merit Award BX004113-01A1 (Hussain, PI) (https://www.research.va.gov/), AHA Postdoctoral Fellowship 19POST34410063 to S.R. (https://www.heart.org/), Academy of Finland 318182 (Ruddock, PI) (https://www.aka.fi/en) and the Wellcome Trust [098051 and 206194] to E. Busch-Nentwich (https://wellcome.ac.uk/). The funders had no role in study design, data collection and analysis, decision to publish, or preparation of the manuscript.

**Competing interests:** The authors have declared that no competing interests exist.

## Author summary

Cardiovascular disease affects about a third of the world's population and is mediated by the buildup of lipids (primarily derived from plasma lipoproteins) in the wall of blood vessels. An essential protein required for the synthesis of the lipoproteins, from insects to humans, is microsomal triglyceride transfer protein (MTP). This protein transfers lipids, including triglycerides and phospholipids, to apolipoprotein B (APOB) for the assembly of ApoB-containing lipoproteins. Here we report a mutation in MTP that blocks the transfer of triglycerides but not phospholipids. Modeling based on the recent crystal structure of the MTP complex suggests the G865V mutation may block access to the region of the protein that binds triglyceride. In zebrafish, the residual phospholipid transfer activity of the mutant protein is sufficient to support secretion of small ApoB-containing lipoproteins and prevent a number of serious health conditions observed in humans harboring null MTP mutations (e.g., intestinal fat malabsorption, growth retardation). These results suggest that selective inhibitors of MTP that can mimic this mutation may be a feasible therapeutic approach to treat dyslipidemias in humans.

## Introduction

From insects to mammals, the bulk transport of hydrophobic lipids through the circulation occurs via lipoproteins [1–3]. In vertebrates, apolipoprotein B-containing lipoproteins (B-lps) are composed of a neutral core of triglycerides (TG) and cholesteryl esters surrounded by a monolayer of phospholipids (PL), free cholesterol, and sphingomyelin. B-lps have one apolipoprotein B (APOB) scaffold protein embedded in the PL monolayer and can also be decorated with other exchangeable apolipoproteins [4, 5]. B-lps are produced by the intestine (chylomicrons) and liver (Very Low Density Lipoproteins (VLDL)) and transport dietary and endogenous lipids and fat-soluble vitamins to the peripheral tissues through the circulation [5–8]. B-lp assembly occurs in the endoplasmic reticulum (ER) and requires the activity of microsomal transfer protein (MTP, human; Mtp, zebrafish) [9–11]. As APOB is translated and translocated into the lumen of the ER, MTP physically interacts with and transfers lipids to APOB to form primordial lipoproteins [10–13]. These nascent lipoproteins are trafficked to the Golgi, where they can be modified and then secreted into the lymph (chylomicrons) or directly into the circulation (VLDL) [14].

MTP is a heterodimer of the large M or MTPα subunit (~97 kDa, encoded by the *MTTP* gene) and the smaller P or MTPβ subunit, protein disulfide isomerase (PDI; ~58 kDa subunit) [15, 16]. Vertebrate MTP can bind and transfer triacylglycerol, diacylglycerol, phospholipid, cholesteryl ester, ceramide, and sphingomyelin between vesicles *in vitro* [17–22]. Kinetic studies suggest that MTP transiently interacts with membranes, acquires lipids, and then delivers these lipids to an acceptor membrane. The transfer of lipids occurs down a concentration gradient and does not require energy [23, 24].

Human mutations in the *MTTP* gene that prevent lipid transfer and APOB secretion cause the disease abetalipoproteinemia (OMIM 200100), characterized by a virtual absence of plasma B-lps [25–28]. Patients exhibit fat malabsorption, intestinal and liver steatosis, low plasma TG and cholesterol levels, and fat-soluble vitamin deficiencies [28–30]. Without adequate supplementation of essential fatty acids and fat-soluble vitamins, these patients can develop a variety of complications including neurological, ophthalmological, and hematological disorders [28, 29].

Although ApoB-containing lipoproteins are critical for lipid transport, elevated numbers of B-lps and high plasma TG and cholesterol concentrations in humans are risk factors for atherosclerosis, cardiovascular disease, and other metabolic diseases [31, 32]. Despite years of research, there are still fundamental knowledge gaps in the factors that regulate lipoprotein production and turnover. Elucidating these molecular details will help identify novel strategies to prevent and treat dyslipidemia. While most of our understanding of the details of B-lp production comes from work in mammalian systems, studying B-lp production in other vertebrates may yield valuable new insights into lipoprotein synthesis and secretion.

In zebrafish and other lecithotrophic teleosts, B-lps are essential during embryonic development to transport lipids from the yolk to the developing tissues before the larvae commence feeding [33–36]. Yolk proteins and lipids are deposited into oocytes by way of vitellogenin, a specialized type of high-density lipoprotein synthesized by the maternal liver [37]. Once in the developing oocyte, vitellogenin undergoes proteolytic cleavage to phosvitin and lipovitellin, and in the case of zebrafish, the lipid is stored in yolk granules/platelets [38, 39]. In zebrafish, the yolk contains many different lipid classes, with the most abundant being cholesterol, phosphatidylcholine, triacylglycerol, phosphatidylinositol, phosphatidylethanolamine, diacylglycerol, cholesteryl esters, and sphingomyelins [40]. During embryogenesis, the lipids in the yolk platelets undergo lipolysis and re-esterification and are then packaged into lipoproteins in the ER of the yolk syncytial layer (YSL), a multi-nucleated cytoplasm that surrounds the yolk mass [41–44]. The zebrafish YSL expresses both ApoB and Mtp and produces large numbers of B-lps [34, 43, 45–48]. These lipoproteins are secreted into the circulation and provide energy and building blocks for the developing embryo.

Zebrafish that are homozygous for a missense mutation (*stalactite*; *mttp*$^{stl/stl}$; L475P) in microsomal triglyceride transfer protein produce very few, small B-lps in the YSL [48]. *mttp*$^{stl/stl}$ mutants exhibit very little lipid in their vasculature and display excessive sprouting angiogenesis as a result of the low circulating levels of ApoB [49, 50]. Notably, these mutants also exhibit morphological changes to their yolk sac, including a more rounded shape and obvious opacity that is not observed in the translucent wild-type embryos [49]. Given that these morphological phenotypes are easy to recognize, we hypothesized that we could identify new modulators of vertebrate B-lp production by screening zebrafish mutant lines for similar defects in yolk morphology. Using this approach, we identified an additional mutant allele of microsomal triglyceride transfer protein, G863V, that we describe in this study.

A phenotypic comparison of the mutant zebrafish *mttp* alleles from embryos to adulthood indicate that the newly identified G863V mutation has a milder effect on B-lp production and lipid malabsorption and no effect on growth when compared to the L475P *stl* mutation. *In vitro* biochemical assays reveal that the *stl* mutation eliminates both the TG and PL transfer activities of Mtp, consistent with all of the analyzed abetalipoproteinemia patient mutations [30]. In contrast, the G863V mutant protein retains PL transfer activity, which allows for continued production of small B-lps, thus preventing intestinal steatosis. These data, together with modeling of the G863V mutation based on the crystal structure of human MTP, provide new insight into the molecular details of lipid transfer by MTP and suggest that selective inhibition of TG transfer by MTP may be a feasible therapeutic approach to treat dyslipidemia.

## Results

### The *c655* allele is a missense mutation in zebrafish *mttp*

A major function of the zebrafish YSL is to rapidly transfer yolk lipids to the developing embryo via B-lps. In order to identify new modulators of B-lp production, we began specifically screening existing zebrafish lines for yolk utilization defects, in particular looking for

embryos with opaque yolks that phenocopy the *mttp^stl/stl* mutant embryos [49] (Fig 1A). Through this approach we discovered mutants with opaque yolks in the background of a zebrafish line carrying an unrelated mutation (*kif7*) (Fig 1A; yolk appears dark when viewed using transmitted light, off-white under incident light, S1 Fig). This opaque yolk phenotype segregates in Mendelian ratios, suggesting the presence of a homozygous recessive mutation. The phenotype was unlinked to the *kif7* genotype and subsequently bred out of the *kif7* line. A Euclidean distance mapping analysis using the Mutation Mapping Analysis Pipeline for

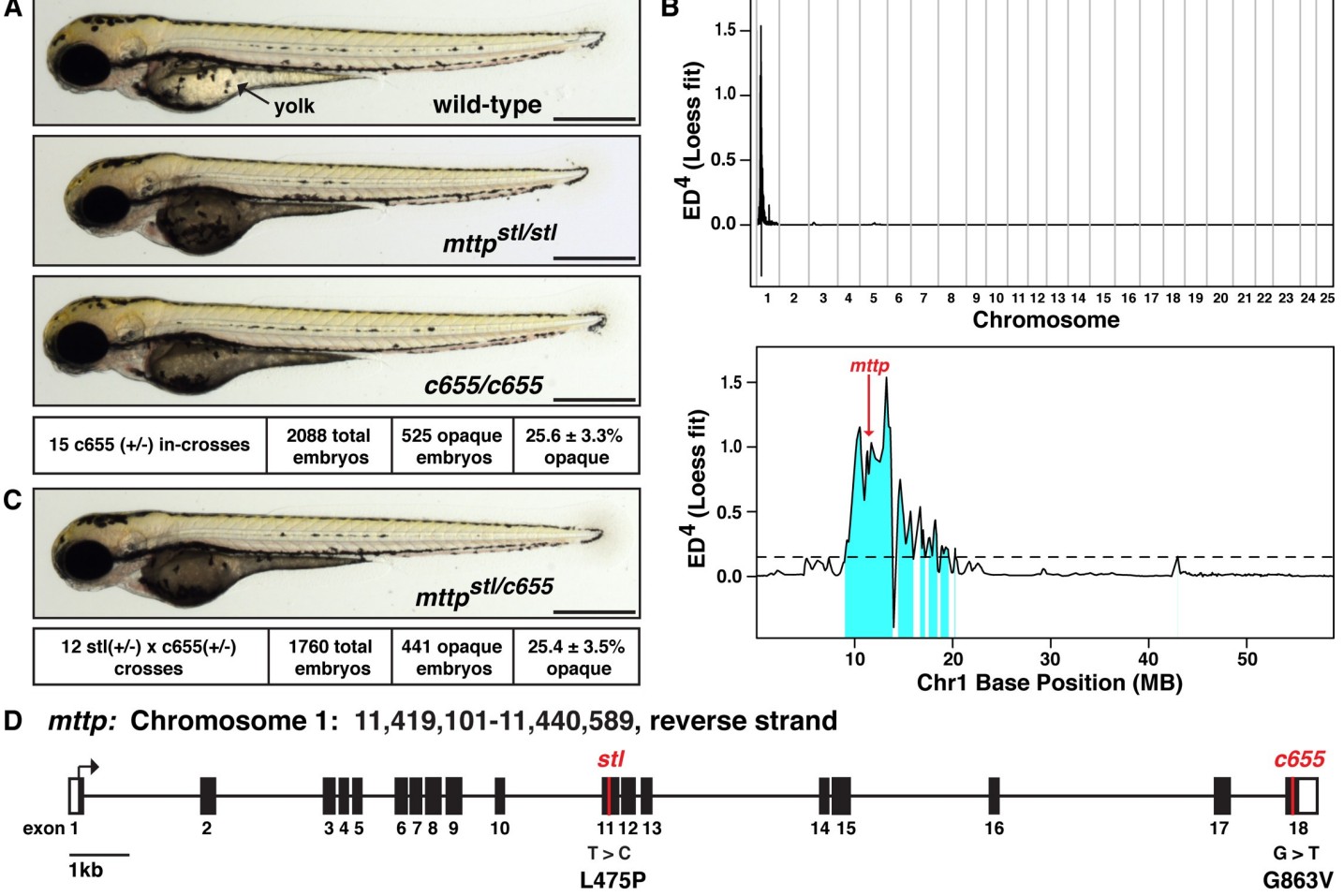

**Fig 1. The *c655* allele is a missense mutation in the M-subunit of microsomal triglyceride transfer protein.** (A) Representative images of a wild-type zebrafish embryo, a homozygous mutant embryo carrying the previously described *stalactite* (*stl*) missense mutation in *mttp*, and a homozygous *c655* mutant embryo at 3 days post fertilization (dpf); Scale = 500 μm. The dark/opaque yolk phenotype in embryos from *c655* heterozygous in-crosses segregated with a Mendelian ratio consistent with a homozygous recessive mutation, mean +/- SD. For source data, see S4 File. (B) Euclidean distance mapping analysis plots produced by MMAPPR [51], showing the likely genomic region of the *c655* mutation. Plot of the LOESS fit to the mapping scores (Euclidean Distance$^4$) across all 25 chromosomes (top) and expanded view of chromosome 1(GRCz10: CM002885.1) (bottom). Single nucleotide variants (SNVs) present in this 11 MB region in *c655* mutant embryos were assessed for their effect on annotated genes using the Ensembl Variant Effect Predictor [52], including using the Sorting Intolerant from Tolerant algorithm (SIFT) [53], to predict the impact of changes on protein-coding sequence (tolerated or deleterious). We extracted variants that alter the protein-coding sequence as candidates for the causal mutation (223 variants in 64 genes, of which 42 are missense variants predicted to be deleterious; S1 File). One of the SNVs linked to the *c655* phenotype was a G>T missense mutation predicted to be deleterious in exon 18 of the microsomal triglyceride transfer protein gene (ENSDARG00000008637, Chr1:11,421,261 GRCz10, red arrow in B shows the position of the G>T missense mutation in *mttp*). (C) Representative image of a trans-heterozygous *mttp^stl/c655* embryo; 3 dpf, scale = 500 μm. The dark/opaque yolk phenotype is present at expected ratios and genotyping confirms that only the embryos with opaque yolks are trans-heterozygous for the *mttp* alleles. (D) Depiction of the *mttp* gene structure highlighting the locations of the *stl* (L475P) (position 11431645 (GRCz10), transcript mtp-204 (ENSDART00000165753.2)) and *c655* (G863V) missense alleles in exon 11 and 18, respectively. An additional SNV in *mttp* at position Chr1:11,421,300 GRCz10 (T>C) causing a missense mutation (M850T) was also identified in *c655* mutants; however, this SNV was not predicted to be deleterious and has been previously noted in Ensembl.

Pooled RNA-seq (MMAPPR) [51], localized the mutation to between 9–20 MB on Chromosome 1 (Fig 1B). One of the single nucleotide variants (SNVs) linked to the opaque yolk phenotype a missense mutation predicted to be deleterious in exon 18 of the *mttp* gene (ENSDARG00000008637, Chr1:11,421,261 GRCz10) (S1 File). This newly identified allele was assigned as Carnegie *c655*.

Complementation crosses between *mttp*^c655/+ heterozygous fish and *mttp*^stl/+ heterozygous fish revealed that the *c655* mutation does not complement the *mttp*^stl mutation (Fig 1C), strongly arguing that the G>T SNV in exon 18 of *mttp* is the causative allele for the *c655* opaque yolk phenotype. This was further confirmed by rescuing the *c655* yolk phenotype with injections of a wild-type *mttp*-FLAG plasmid at the 1-cell stage (S2 Fig). No change in mRNA expression was noted for *mttp* in the *mttp*^c655/c655 mutants in our RNA-seq data-set ($\log_2$[fold change] = 0.18, adj. $p$ = 0.19).

Both the *mttp*^stl allele and *mttp*^c655 allele are missense mutations. The *stl* allele results in the conversion of a leucine to a proline at residue 475 (L475P) and the *c655* mutation is a glycine to valine mutation in the C-terminus of the protein at residue 863 (G863V) (total length = 884 residues) (Fig 1D). Although the *mttp*^stl/stl and *mttp*^c655/c655 fish both exhibit opaque yolks, the *mttp*^stl/stl mutants exhibit a more severe phenotype than the *mttp*^c655/c655 mutants, in that their yolks are darker, more rounded, and they retain the opaque phenotype longer during development (Fig 1A, S3 Fig). This difference in phenotype suggested that the two missense mutations may be affecting Mtp protein function differentially.

## Yolk opacity is due to the aberrant accumulation of cytoplasmic lipid droplets in the yolk syncytial layer

While yolk opacity had been observed in the original description of the *mttp*^stl/stl mutants, the etiology was never explained [49]. When MTP is mutated or absent, B-lp production is reduced or absent and TG accumulates in cytoplasmic lipid droplets (LDs) [54–56]. We have previously shown that accumulation of LDs in intestinal enterocytes of zebrafish larvae fed a high-fat meal causes the gut to be opaque [57] (S4 Fig), most likely due to the lipid droplets' ability to scatter light [58, 59]. Therefore, we hypothesized that the yolk opacity in the *mttp* mutant embryos is due to aberrant accumulation of LDs in the cytoplasm of the YSL.

Using transmission electron microscopy, we found that the YSL in the wild-type embryos contains very few, if any, canonical LDs, whereas the *mttp*^stl/stl, *mttp*^c655/c655, and trans-heterozygous *mttp*^stl/c655 embryos accumulate substantial numbers of cytoplasmic LDs (Fig 2A and 2B). LDs in *mttp*^stl/stl mutants are more numerous and more uniform in size, whereas the *mttp*^c655/c655 mutants often had very large LDs in addition to small droplets (Fig 2C). As a result, the number of LDs per area of the YSL is reduced in the *mttp*^c655/c655 mutants compared to *mttp*^stl/stl mutants (Fig 2D). The trans-heterozygous fish had LDs that were more similar in size to the *mttp*^stl/stl mutants and had a trend toward fewer lipid droplets per YSL area, although this was not significant. The differences in the concentration and size of LDs between the mutants may result in differential effects on the degree of light scattering, which could explain the differences in opacity noted between mutants (Fig 1A, S3 Fig). These data also confirm that yolk opacity is a readily visible phenotype for perturbations in embryonic lipid flux.

## *c655* mutants secrete more lipoproteins from the YSL than *stl* mutants

In recent work, we have shown that *mttp*^stl/stl mutant embryos produce very few B-lps, which are much smaller than lipoproteins produced by wild-type embryos [48]. To examine how the number and size of B-lps is affected by the *c655* mutation, we crossed the *mttp*^c655 mutation into our LipoGlo reporter line. These fish express an in-frame fusion of the luciferase reporter

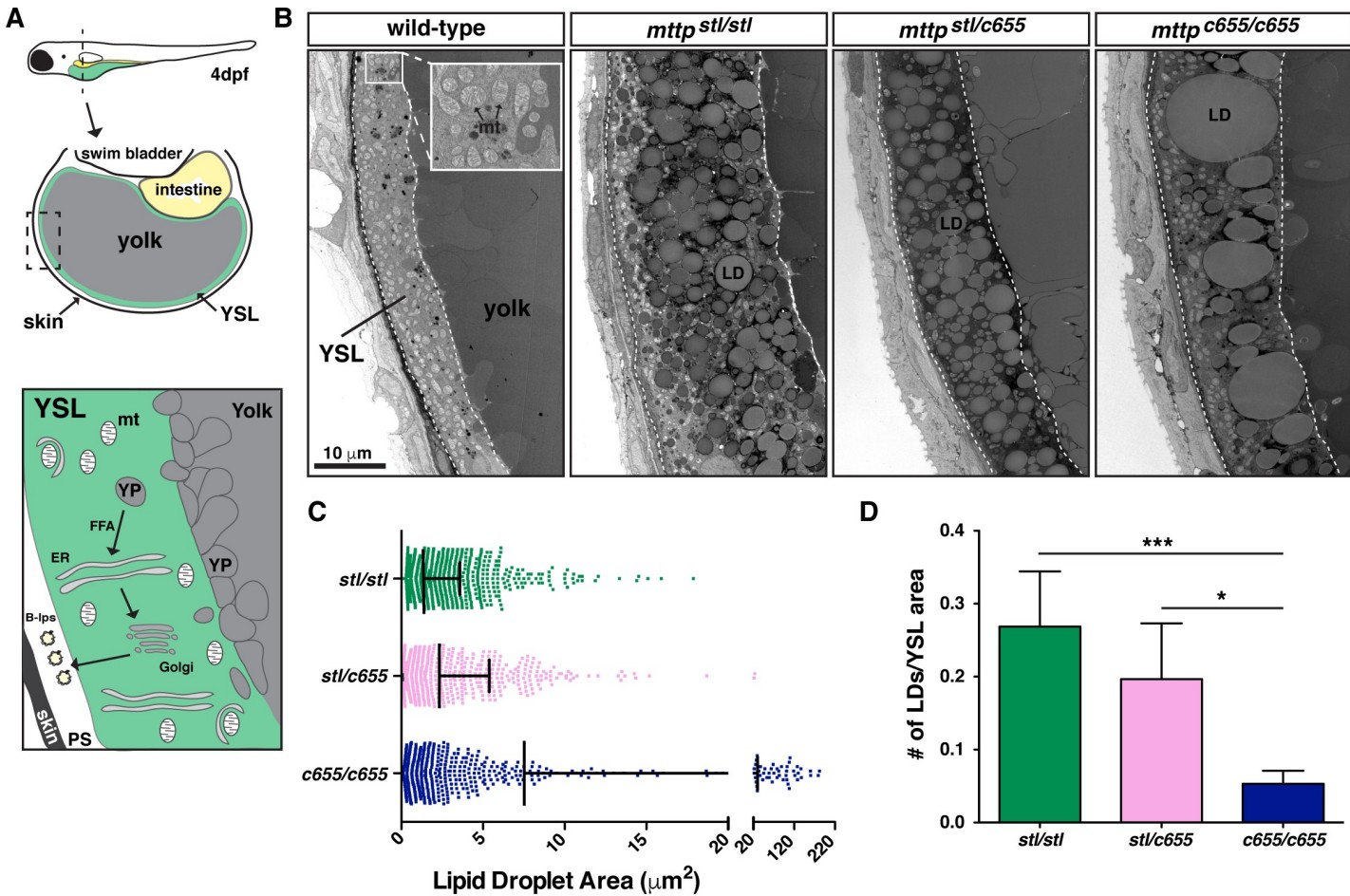

**Fig 2. The opaque yolk phenotype results from the accumulation of aberrant cytoplasmic lipid droplets in the yolk syncytial layer.** (A) (Top) Cartoon depicting the cross-sectional view of a 4 dpf zebrafish embryo. The YSL surrounds the yolk mass and serves as the embryonic digestive organ. The dashed box indicates the view expanded in the bottom panel and in panel B. (Bottom) Stored yolk lipids undergo lipolysis in yolk platelets (YP), presumably releasing free fatty acids into the YSL. These fatty acids are re-esterified in the ER bilayer to form TG, PL, and cholesterol esters. The lipids are packaged into B-lps in the ER with the help of Mtp and are likely further processed in the Golgi before being secreted into the perivitelline space (PS) and then circulation. (B) Representative transmission electron micrographs of the yolk and YSL from wild-type and *mttp* mutants; dashed lines delineate the YSL region, mt = mitochondria, scale = 10 μm. (C) Quantification of lipid droplet size in *mttp* mutants, n ≥ 700 lipid droplets in 2 fish per genotype; mean +/- SD. (D) Quantification of the number of lipid droplets per YSL area, n = 7–9 YSL regions per genotype (3–5 regions per fish, 2 fish per genotype); mean +/- SD, Kruskall-Wallis with Dunn's Multiple Comparison test, vs. *mttp^{c655/c655}*, * $p < 0.05$, *** $p < 0.001$.

NanoLuc at the C-terminus of the apolipoprotein Bb.1 gene (Fig 3A, S5 Fig). Since ApoB is an obligate structural component of B-lps with only one copy per lipoprotein particle [60], the relative number and size of tagged lipoprotein particles can be quantified in extracts from transgenic fish using the LipoGlo assays as previously described [48].

B-lp levels were measured in whole fish lysate throughout embryonic development from 2–6 days post fertilization (dpf). During this time, the fish rely solely on yolk lipids as their digestive system is not fully developed until 5 dpf [61] and no exogenous food was provided. Because ApoB is primarily expressed in the YSL prior to 5 dpf [45], the ApoB quantity measurements largely reflect YSL-derived B-lps. Wild-type embryos exhibit an increase in B-lp particle number from 2–3 dpf as yolk lipid is packaged into lipoproteins. Subsequently, numbers decline as the yolk is depleted, the lipids in the B-lps are taken up by target tissues, and lipoprotein particles are degraded (Fig 3B). As we have shown previously, *mttp^{stl/stl}* embryos have profound defects in B-lp production (Fig 3B) [48]. In contrast, *mttp^{c655/c655}* embryos have

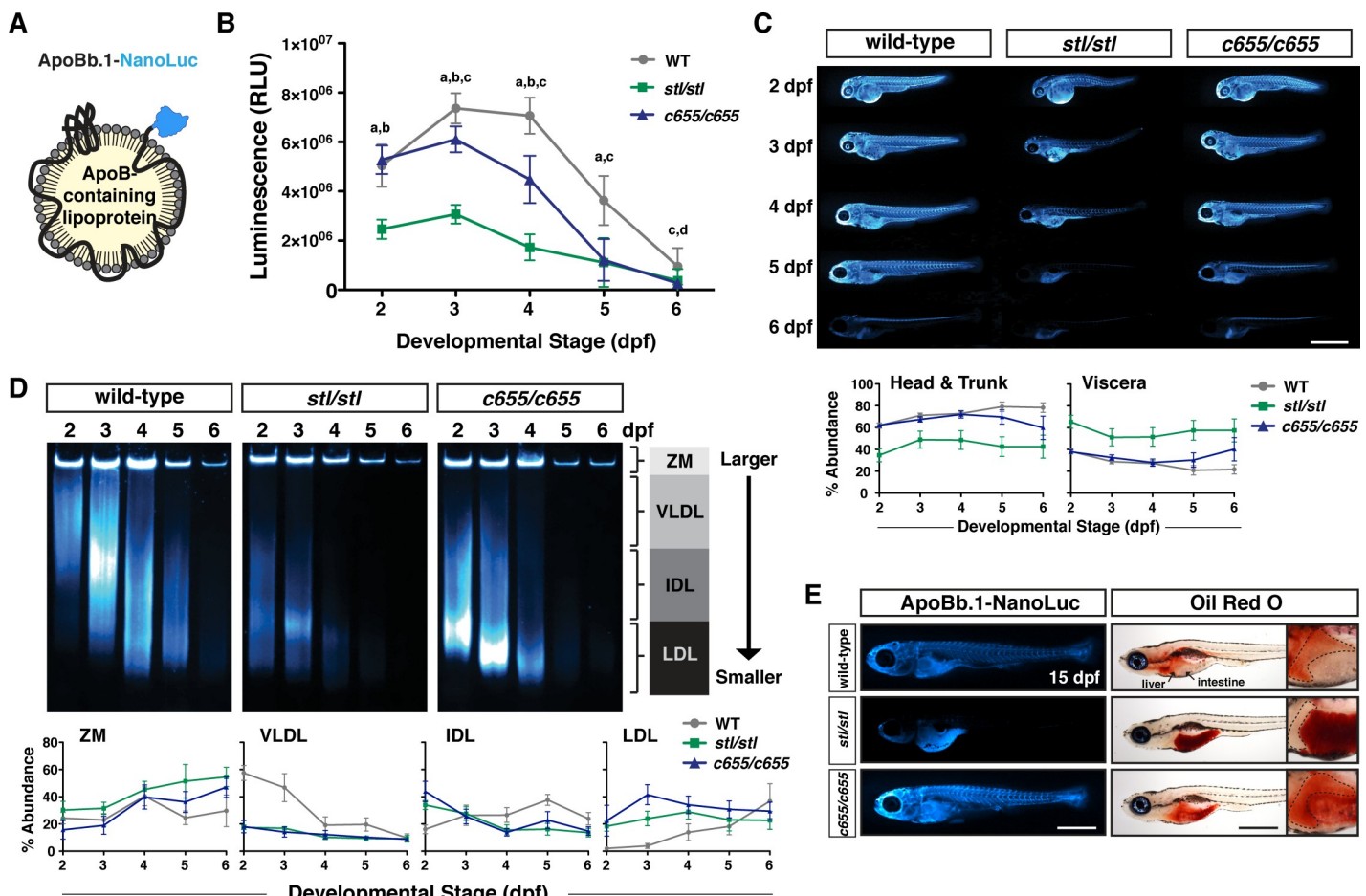

**Fig 3. The *c655* mutation supports secretion of small LDL-sized lipoproteins *in vivo*.** (A) LipoGlo fish express the NanoLuc luciferase enzyme as a C-terminal fusion on ApoBb.1 as a result of TALEN-based genomic engineering [48]. (B) LipoGlo signal (RLU: relative luminescence units) in WT, *mttp^{stl/stl}*, and *mttp^{c655/c655}* fish throughout embryonic development (2–6 dpf). Results represent pooled data from 3 independent experiments, n = 22–34 fish/genotype/time-point. Significance was determined with a Robust ANOVA, Games-Howell post-hoc tests were performed to compare genotypes at each day of development, and p-values were adjusted to control for multiple comparisons, a = WT vs. *mttp^{stl/stl}*, $p < 0.001$, b = *mttp^{c655/c655}* vs. *mttp^{stl/stl}*, $p < 0.001$, c = WT vs. *mttp^{c655/c655}*, $p < 0.001$, d = WT vs. *mttp^{stl/stl}*, $p < 0.05$. (C) Representative whole-mount images of B-lp localization using LipoGlo chemiluminescent microscopy in WT, *mttp^{stl/stl}*, and *mttp^{c655/c655}* fish throughout development; scale = 1 mm. Graphs represent pooled data from 3 independent experiments, n = 13–19 fish/genotype/time-point; *mttp^{stl/stl}* had a significantly different ApoB localization from WT and *mttp^{c655/c655}*, $p < 0.001$, Robust ANOVA. Games-Howell post-hoc analysis reveals statistical differences at all developmental stages; $p < 0.05$–0.001. (D) Representative LipoGlo PAGE gels and quantification of B-lp size distribution from whole embryo lysates during development. B-lps are divided into four classes based on mobility, including zero mobility (ZM) and three classes of serum B-lps (VLDL, IDL, and LDL). Graphs show subclass abundance for WT, *mttp^{stl/stl}*, and *mttp^{c655/c655}* fish at each day of embryonic development as described in [48]. Results represent pooled data from n = 9 samples/genotype/time-point; at each particle class size, there were statistically significant differences between genotypes (Robust ANOVA, $p < 0.001$). Games-Howell post-hoc analysis revealed numerous differences between genotypes at each developmental stage, see S7 Fig. (E) Representative whole-mount images of LipoGlo microscopy and Oil Red O imaging in 15 dpf embryos chow-fed for 10 days and fasted ~18 h prior to fixation; scale = 1 mm. Livers (outlined) are magnified for clarity in insets on right. Results represent pooled data from 3 independent experiments, n = 15 fish/genotype/time-point.

the same relative number of ApoB particles as wild-type embryos at 2 dpf, but from 3–6 dpf the numbers of particles never reach wild-type levels and decline more rapidly (Fig 3B).

To assess the localization of the B-lps throughout the embryos during development, we fixed the embryos expressing ApoBb.1-NanoLuc and performed chemiluminescent whole-mount imaging (Fig 3C). We found that *mttp^{c655/c655}* embryos exhibit a similar distribution pattern of LipoGlo to wild-type embryos throughout 2–4 dpf, but consistent with the quantitative assay, the signal in the head and trunk decline more rapidly in *mttp^{c655/c655}* fish. By 6 dpf, both wild-type and *mttp^{c655/c655}* fish show an accumulation of ApoB in the liver and the spinal

cord (Fig 3C) [48]. In contrast, ApoB in the *mttp*<sup>stl/stl</sup> embryos is predominantly localized to the YSL/viscera at all stages and is present at very low levels throughout the rest of the body (Fig 3C) [48].

## *c655* mutants only produce small lipoproteins

To examine whether the *c655* mutation alters the size distribution of B-lps, we performed native polyacrylamide gel electrophoresis of larval homogenates expressing the LipoGlo reporter. Following electrophoretic separation and chemiluminescent imaging of the gels, B-lps were classified into four different classes based on their migration distance (zero mobility (ZM), very low-density lipoproteins (VLDL), intermediate-density lipoproteins (IDL), or low-density lipoproteins (LDL)) [48]. During development, the B-lp profile in wild-type embryos is initially dominated by VLDL (2 dpf), but expands to include IDL and LDL by 3–4 dpf as the VLDL particles produced by the YSL are lipolyzed by circulating lipases throughout the body (Fig 3D; S6 Fig, S7 Fig) [48]. By 5–6 dpf, the yolk is depleted; no additional large VLDL particles are produced, leaving only small LDL particles. As we have shown previously, the *mttp*<sup>stl/stl</sup> embryos predominantly produce small B-lps (Fig 3D, 2 dpf) [48]. Similarly, *mttp*<sup>c655/c655</sup> embryos also produce very few VLDL particles (Fig 3D, 2 dpf), and instead produce predominantly IDL and LDL-sized particles.

## *c655* mutants secrete lipoproteins more effectively from the intestine than *stl* mutants

To test the hypothesis that *mttp*<sup>c655/c655</sup> mutants are able, like wild-type fish, to secrete B-lps from the intestine, we performed chemiluminescent imaging using the LipoGlo reporter in 15 dpf larvae fed a chow diet for 10 days and then fasted overnight. Wild-type LipoGlo fish have abundant ApoB throughout their circulation and tissues (73.1 +/- 4.0% in head and trunk vs. 26.9 +/- 4.0% in viscera, mean +/- SD, n = 15 fish) (Fig 3E). Similar to what was noted in the embryos, *mttp*<sup>stl/stl</sup> fish have abundant LipoGlo signal in their intestine and much less in other tissues compared to WT (41 +/- 11% in head and trunk vs. 59 +/- 11% in viscera, $p < 0.001$, Kruskall-Wallis & Dunn's Multiple Comparisons Test) (Fig 3E). In contrast, the *mttp*<sup>c655/c655</sup> mutation does not prevent secretion of ApoB to the body tissues (73.1 +/- 3.7% in head and trunk vs. 26.9 +/- 3.7% in viscera). Consistent with this observation, staining the neutral lipids with Oil Red O indicates that *mttp*<sup>stl/stl</sup> mutants retain substantial lipid in their intestines, whereas *mttp*<sup>c655/c655</sup> mutant fish have less lipid remaining in their intestines, but do accumulate some lipid in their livers (Fig 3E). These data argue that the *stl* mutation severely reduces B-lp secretion, not only from the yolk in embryos, but also from the intestinal enterocytes in larvae, whereas the *c655* mutation only mildly decreases ApoB secretion in both embryos and larvae.

## *c655* mutants do not exhibit growth defects

Patients with abetalipoproteinemia often present in infancy with fat malabsorption, growth retardation, diarrhea, and failure to thrive (reviewed in [29]), and whole body deficiency of MTP in a murine model is embryonic lethal [56]. While the original description of zebrafish *mttp*<sup>stl/stl</sup> mutants noted that the fish did not survive past 6 dpf [49], we found that some of the *stl* mutants not only survive past early larval stages (Fig 3E), but can live to be at least 2 years old. However, these fish are generally much smaller in length and mass (Fig 4A, S8 Fig) and their viability is reduced relative to their siblings (expected 25%, observed 3.8% [5/131 fish] at 7.5 mo). Survival rates are better when the mutants are reared separately and are not competing with wild-type and heterozygous siblings for food. Although these fish can reproduce, this

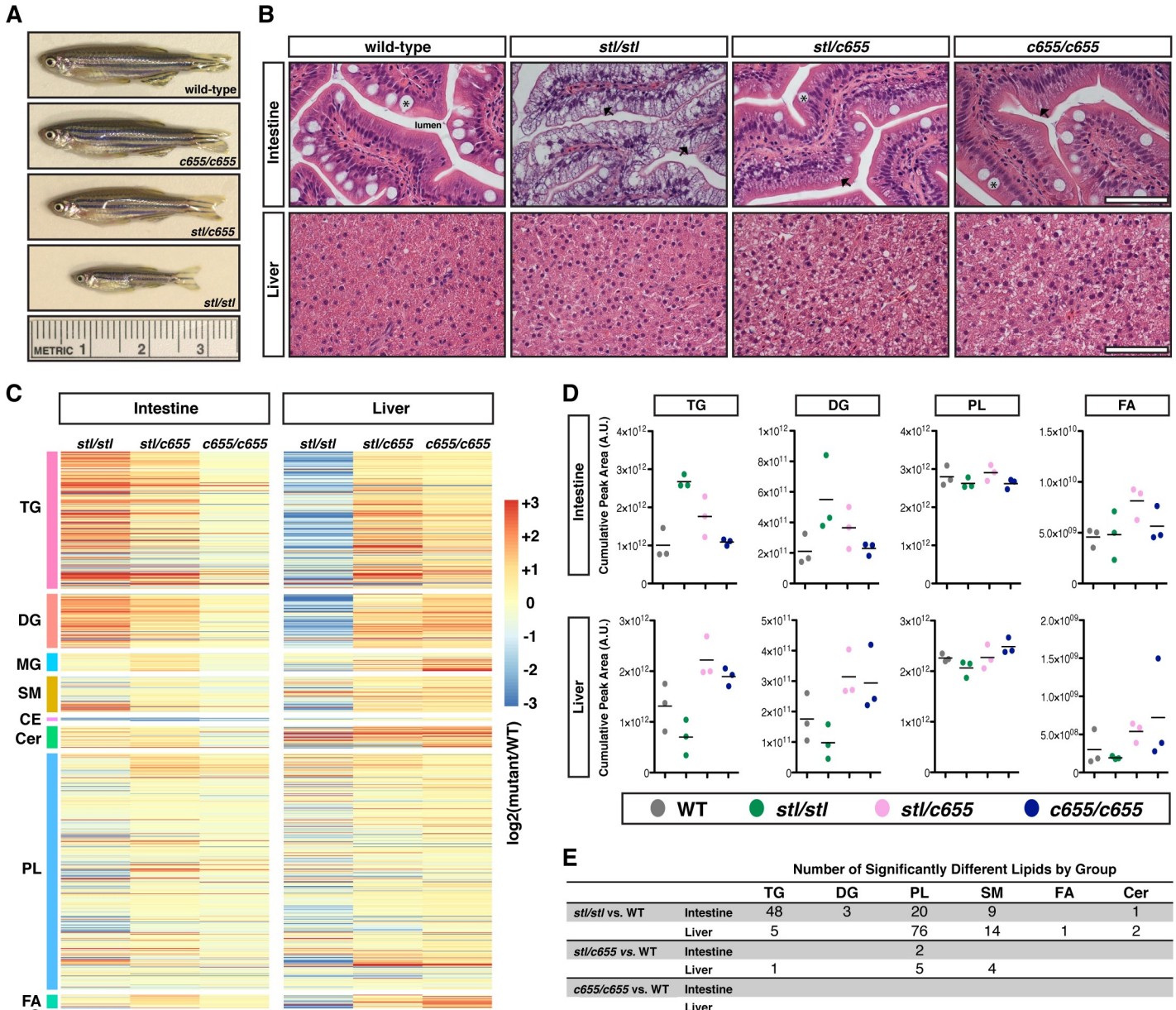

**Fig 4. The *stl* and *c655 mttp* mutations have differential effects on growth and the accumulation of lipid in intestine and liver.** (A) Representative images of male WT and *mttp* mutant fish at 12 weeks of age. (B) Representative images of H&E stained intestine and liver from adult male WT and *mttp* mutant fish (7.5 mo), scale = 50 μm, * indicate goblet cells, arrows indicate representative lipid accumulation in enterocytes. (C–E) Intestine and liver tissue from adult male fish were extracted based on equal concentration of protein. Tissue lipid extracts from WT and *mttp* mutant fish were quantitated using an HPLC system coupled to a tandem mass spectrometer (LC-MS/MS) (n = 3; 1 fish per sample/genotype). (C) Heat maps represent fold-change from WT of over 1000 individual lipid species grouped into lipid classes (triacylglycerol [TG, n = 274], diacylglycerol [DG, n = 108], monoacylglycerol [MG, n = 36], sphingomyelin [SM, n = 72], cholesterol ester [CE, n = 7], ceramides [Cer, n = 44], phospholipid [PL, n = 472], free fatty acid [FA, n = 27] and other lipids [O; including sterols, sphingosine, sulfatide, zymosteryl and wax esters, n = 10]). (D) Quantification of total intestinal and liver TG, DG, PL, and FA from mutant lines as expressed as a sum of lipid group (n = 3). For additional lipid groups, see S11 Fig. (E) The number of individual lipid species data from panel (C) that are statistically different from WT (adj. *p* < 0.20).

is rare. We hypothesize that during the maintenance of this mutant line since its original characterization, a modifier has been eliminated that, when present in the *stl* background, was incompatible with life. In support of this hypothesis, the excessive sprouting angiogenesis defect, for which the *stalactite* mutation was named [49], was also not as severe as originally

described (S9 Fig). Whether the proposed modifier directly affects the secretion of B-lps, or some other aspect of development, is currently unclear. In stark contrast, the $mttp^{c655/c655}$ mutants do not exhibit reduced viability (expected 25%, observed 21.3% [36/169 fish] at 7.5 mo), and we did not find any reduction in size or fertility of the *c655* mutants compared with siblings (Fig 4A, S8 Fig). No difference in length or mass was also noted in fish trans-heterozygous for $mttp^{stl/c655}$ (Fig 4A, S8 Fig), suggesting one copy of $mttp^{c655}$ is sufficient for normal growth.

## *c655* mutant adults are largely protected from intestinal steatosis

The LipoGlo and Oil Red O imaging of larvae at 15 dpf (Fig 3E) suggested that $mttp^{c655/c655}$ mutants are more effective at packaging dietary lipids into chylomicrons than $mttp^{stl/stl}$ mutants. To assess whether this is also true in adult fish, we did hematoxylin & eosin (H&E) staining of intestinal tissue from fasted adults. Consistent with our findings at 15 dpf, there is gross accumulation of lipid in the cytoplasm of enterocytes in the $mttp^{stl/stl}$ fish (Fig 4B, S10 Fig), but the $mttp^{c655/c655}$ mutants were largely protected from this abnormal lipid retention. Trans-heterozygous fish exhibited an intermediate phenotype. Quantification of lipids using LC-MS/MS is consistent with the tissue histology observed in *mttp* mutants. Lipids from $mttp^{c655/c655}$ intestine are largely indistinguishable from WT, whereas $mttp^{stl/stl}$ intestines have approximately 3-fold more TG (Fig 4C–4E, S11 Fig, S2 File & S3 File). These data suggest that the growth defects observed in $mttp^{stl/stl}$ mutants result from defects in dietary lipid absorption in the intestine. However, the residual Mtp activity in the $mttp^{c655/c655}$ mutant fish is sufficient to prevent intestinal steatosis and promote normal growth.

Besides accumulating lipids in the intestine, abetalipoproteinemia patients can also develop hepatic steatosis (reviewed in [29]). Similarly, hepatocyte-specific deficiency of MTTP in mice causes TG and cholesterol to accumulate in the liver [54, 55]. H&E staining and lipid quantification by LC-MS/MS of liver tissue from *mttp* fish mutants was performed to examine the level of steatosis. While $mttp^{c655/c655}$ liver lipid content was also indistinguishable from WT, we were surprised that the $mttp^{stl/stl}$ mutants exhibited little histological or biochemical evidence of hepatic lipid accumulation (Fig 4B–4E, S11 Fig, S2 File & S3 File). However, these data are in agreement with findings that combined intestinal and liver deficiency of MTTP in mice results in accumulation of TG in the intestine, but not in the liver [22].

## The *c655* mutation in zebrafish *mttp* disrupts TG transfer activity but not PL transfer activity of the Mtp complex

The dissimilar phenotypes of B-lp secretion between the *stl* and *c655* mutations *in vivo* strongly suggest that the two mutations are differentially affecting Mtp function. To investigate how each of the mutations affects Mtp function, we turned to cell- and *in-vitro*-based assays. First, to confirm the differences in ApoB secretion that were noted *in vivo*, COS-7 cells expressing human APOB48 were co-transfected with either an empty vector (pcDNA3) or a vector containing wild-type zebrafish *mttp*, $mttp^{stl}$, or $mttp^{c655}$, all with a C-terminal FLAG-tag. Consistent with our findings *in vivo*, COS-7 cells expressing stl-FLAG protein secreted very little APOB48 into the media (Fig 5A), causing retention of APOB48 inside the cells (Fig 5B). In contrast, the c655-FLAG-expressing cells were still able to secrete APOB48 into the media, albeit with reduced efficiency compared to wild-type Mtp-FLAG (Fig 5A and 5B), again confirming the data in fish.

To understand mechanistically how the two mutations differentially alter APOB48 secretion, we performed additional assays. First, to determine whether the mutant proteins are localized properly in the ER, transfected COS-7 cells were immunostained using an anti-

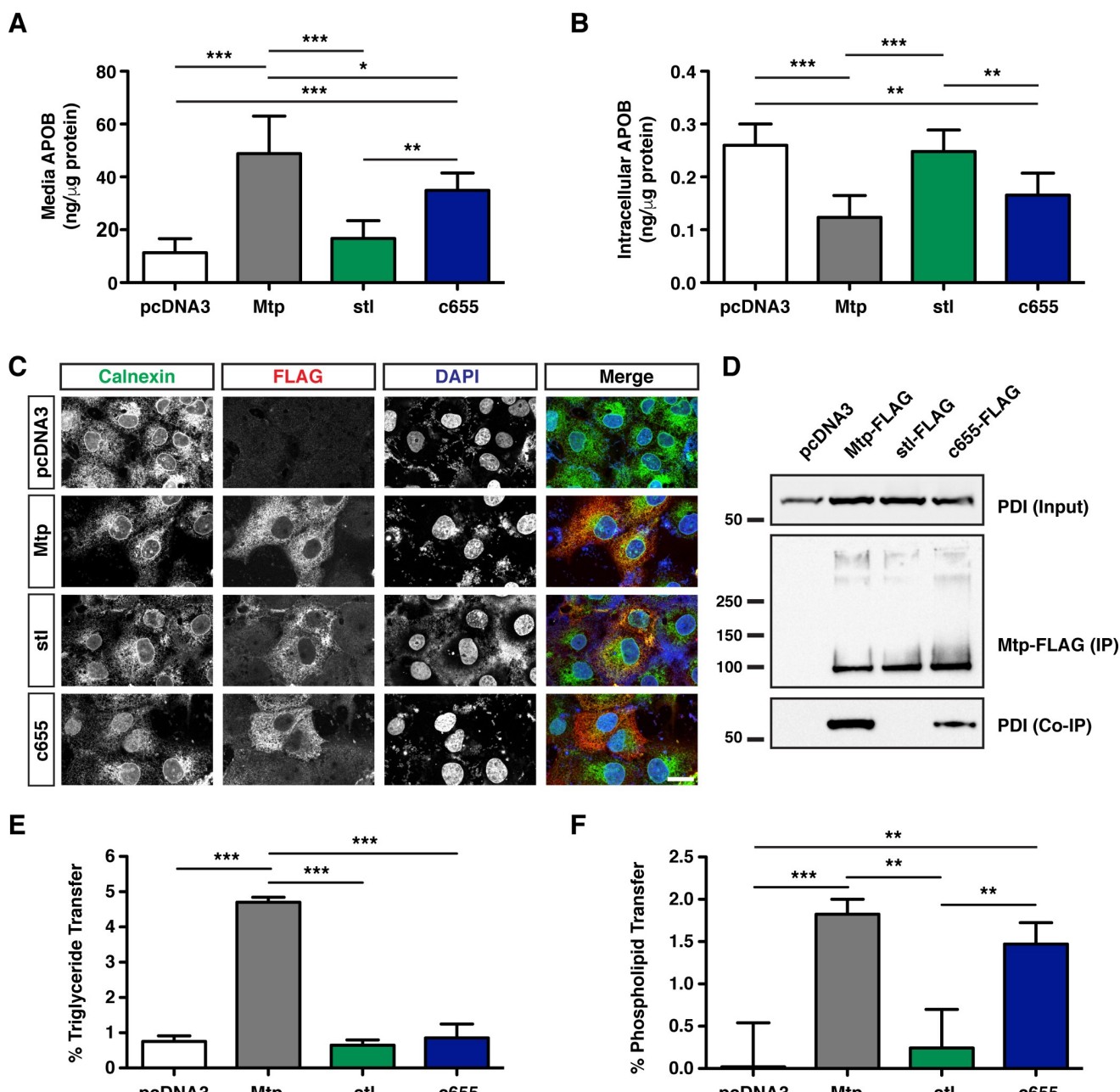

**Fig 5. The *c655* mutation disrupts TG transfer activity, but not PL transfer activity of the zebrafish Mtp complex.** (A, B) COS-7 cells were first transfected with an expression vector for human APOB48 (5 μg), distributed equally in 6-well plates, and subsequently transfected with plasmids expressing either wild-type zebrafish *mttp*-FLAG, *mttp*^*stl*^-FLAG, *mttp*^*c655*^-FLAG, or empty vector (pcDNA3) (3 μg). After 72 h, APOB48 was measured via ELISA in media (A) or in the cell (B). Data are representative of 7 independent experiments (each data point is the mean of three technical replicates), mean +/- SD, One-Way ANOVA with Bonferroni post-hoc tests, * $p < 0.05$, ** $p < 0.01$, *** $p < 0.001$. (C) Representative immunofluorescent staining using anti-FLAG (red) and anti-Calnexin (green) antibodies in COS-7 cells expressing wild-type or mutated *mttp*-FLAG constructs; scale = 25 μm. The percentage of cells expressing the FLAG-tagged proteins was similar among all groups (Mtp-FLAG 37%, stl-FLAG 31%, c655-FLAG 41% transfection efficiency). (D) Zebrafish Mtp-FLAG proteins (WT, stl and c655) were immunoprecipitated from COS-7 cell lysate (400 μg) using the M2 flag antibody and immunoblots were probed for both FLAG and PDI (Representative of 2 experiments). For input, 15 μg of cell lysate was used. (E) COS-7 cells were transfected with plasmids expressing pcDNA3, wild-type zebrafish *mttp-FLAG*, or mutant *mttp*-FLAG constructs. Cells were lysed and 60 μg of protein was used to measure the % TG transfer of nitrobenzoxadiazole (NBD)-labeled triolein from donor to acceptor vesicles after 45 min; n = 3 (each n is the mean of three technical replicates from independent experiments), mean +/- SD, One-way ANOVA with Bonferroni post-hoc tests, *** $p < 0.001$. (F) Wild-type and mutant Mtp proteins were purified using anti-FLAG antibodies and used to measure the % transfer of NBD-labeled phosphoethanolamine triethylammonium from donor to acceptor vesicles after 180 min; n = 3 (each n is the mean of three technical replicates from independent experiments), mean +/- SD, randomized block ANOVA with Bonferroni post-hoc tests, *** $p < 0.001$.

FLAG antibody. Immunostaining shows that both zebrafish stl and c655 mutant proteins are located in the ER, as shown by co-localization with the ER-marker calnexin (Fig 5C). To assess whether the mutated M subunits are interacting properly with PDI to form a complex, we performed co-immunoprecipitation and western blot analyses. The FLAG-tagged proteins were immunoprecipitated with anti-FLAG antibodies (Fig 5D) and precipitated samples were probed for PDI by immunoblotting. Despite its localization in the ER, the stl mutant protein did not co-immunoprecipitate with PDI (Fig 5D). In contrast, the c655 mutant protein did associate with PDI but to a lesser extent than wild-type Mtp (Fig 5D).

Based on these results, we hypothesized that the *stl* mutation would eliminate all lipid transfer activity because the Mtp complex was not intact, and that the *c655* mutation would cause a reduction in lipid transfer, consistent with the smaller lipoproteins noted in the *mttp^c655/c655* mutant embryos. To test this idea, we performed TG and PL transfer assays *in vitro* using cell lysates. As expected, the stl-FLAG mutant protein showed virtually no TG transfer activity or transfer of a fluorescent phosphatidylethanolamine (PE) analog when compared to wild-type zebrafish Mtp (Fig 5E and 5F, S12 Fig). However, we were surprised to find that unlike all previously identified hypomorphic human *MTTP* alleles, the *c655* mutation has differential effects on the transfer activities of different lipid species. TG transfer was abolished, but PL transfer activity was only decreased by ~20% compared to wild-type Mtp (Fig 5E and 5F, S12 Fig). These studies suggested that the attenuated B-lp secretion observed in *mttp^stl/stl* mutants might result from its failure to interact with PDI. In contrast, the *mttp^c655/c655* mutant is able to support more B-lp assembly because it retains PL transfer activity.

## Generating the corresponding *c655* mutation in human MTTP (G865V) also disrupts TG transfer activity but not PL transfer activity

The glycine residue (G863) mutated in *mttp^c655/c655* fish is conserved in human MTTP (G865). To determine the effects of the *c655* mutation on lipid transfer activities of human MTP, we repeated the cell- and *in-vitro*-based assays with FLAG-tagged wild-type and mutated G865V human MTP proteins. Both the wild-type and MTP G865V mutant proteins localize to the ER (Fig 6A). However, a reduced interaction between PDI and the G865V mutant form of the M subunit was observed (Fig 6B). Similar to our findings with the zebrafish proteins, the human G865V mutation reduced but did not prevent the secretion of APOB48 from COS-7 cells (Fig 6C and 6D). The G865V mutation inhibited TG transfer activity to levels comparable to treatment of the wild-type hMTP protein with the MTP inhibitor lomitapide (MTTPi) (Fig 6E). However, the G865V mutated protein retained ~80% of PL transfer activity (Fig 6F, S13 Fig), similar to what was noted for the zebrafish G863V mutation (Fig 5F). This remaining activity was abolished by treatment with lomitapide (Fig 6F).

## Structural analysis of MTP mutations

The MTP M subunit has three major structural domains: an N-terminal half beta-barrel, a middle alpha-helical domain, and a C-terminal domain consisting of two beta-sheets and two alpha-helices that encompasses the lipid-binding site [16]. The amino acid sequence of the zebrafish Mtp M subunit is 54% identical to that of the human protein, while the PDI P subunits are ~75% identical. Homology modeling based on the crystal structure of human MTP (PDB ID: 6I7S) indicates that the predicted tertiary and quaternary structures are highly conserved (Fig 7A).

The leucine residue mutated in the *mttp^stl/stl* mutant fish (L475P) is also found in human MTTP (L477) and lies within a highly conserved stretch of amino acids located in helix 10 of the alpha-helical domain of the M subunit (Fig 7A, 7B and 7C). This residue does not interact

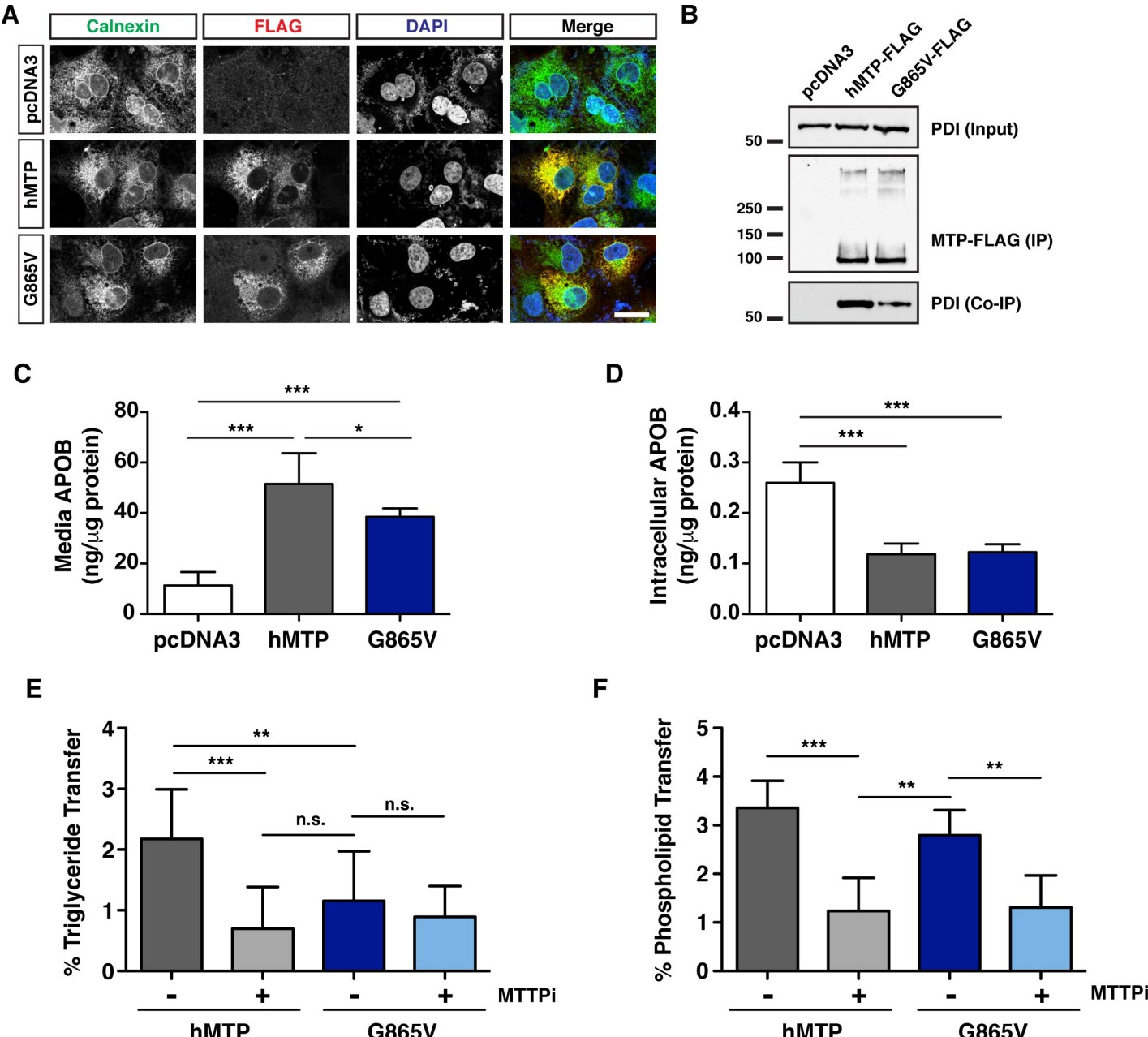

**Fig 6. The corresponding *c655* mutation in human MTTP disrupts TG transfer but not PL transfer activity.** (A) Immunofluorescence in COS-7 cells expressing wild-type human MTTP-FLAG or human MTTP(G865V)-FLAG proteins using anti-FLAG (red) and anti-Calnexin (green) antibodies; scale = 25 μm. (B) Human MTP-FLAG proteins (WT and G865V) were immunoprecipitated from COS-7 cell lysate (400 μg protein) using the M2 flag antibody and immunoblots were probed for both FLAG and PDI. For input, 15 μg protein was used. (C, D) COS-7 cells were co-transfected with human APOB48 and either wild-type human *MTTP*-FLAG, *MTTP*(G865V)-FLAG or empty pcDNA3 plasmids. After 72 h, APOB48 was measured via ELISA in media (C) or in the cell (D). Data are representative of 7 independent experiments (each data point is the mean of three technical replicates), pcDNA3 control data is re-graphed from Fig 5A & 5B (data for Figs 5A, 5B, 6C and 6D were generated together); mean +/- SD, One-Way ANOVA with Bonferroni post-hoc tests, * $p < 0.05$, *** $p < 0.001$. (E) COS-7 cells were transfected with plasmids expressing human wild-type or *MTTP*(G865V)-FLAG constructs. Cells were lysed and 60 μg of protein was used to measure TG transfer activity in the presence or absence of the MTP inhibitor lomitapide (MTTPi, 1 μM) (% after 45 min); n = 3 (each n is the mean of three technical replicates from independent experiments), mean +/- SD, One-way ANOVA with Bonferroni post-hoc tests, ** $p < 0.01$, *** $p < 0.001$, n.s. not significant). (F) Wild-type and mutant MTP proteins were purified using anti-FLAG antibodies and used to measure PL transfer in the presence or absence of lomitapide (MTTPi, 1 μM) (180 min); n = 3 (each n is the mean of three technical replicates from independent experiments), mean +/- SD, randomized block ANOVA with Bonferroni post-hoc tests, ** $p < 0.01$, *** $p < 0.001$.

with PDI and does not face the lipid-binding site. However, it is involved in packing helix 10 with helices 9 and 11, while a neighboring residue (N477 in zebrafish, N479 in human) forms hydrogen bonds with the backbone of Q698/Q700 in the C-sheet which forms the lipid-binding site. The leucine to proline mutation is likely to disrupt helix 10 and the packing of the alpha-helical domain against the lipid-binding domain, thereby affecting lipid transfer activity indirectly. This is consistent with reported mutations in this region of the alpha-helical domain that cause abetalipoproteinemia, including L435H, Y528H and S590I (Reviewed in [30]), affecting lipid transfer activity indirectly by inducing conformational changes and/or destabilizing the structure [16, 62, 63]. Our immunoprecipitation data indicating that the L475P mutant protein fails to bind PDI also suggests that this mutation may be destabilizing the structure (Fig 5D).

The glycine residue mutated in the C-terminus of *mttp*[c655/c655] mutants (G863V) is also conserved in the human sequence (G865) (Fig 7A and 7B). This residue is situated at one of the contact points between the M subunit and PDI; it lies within 4Å of the catalytic site in the

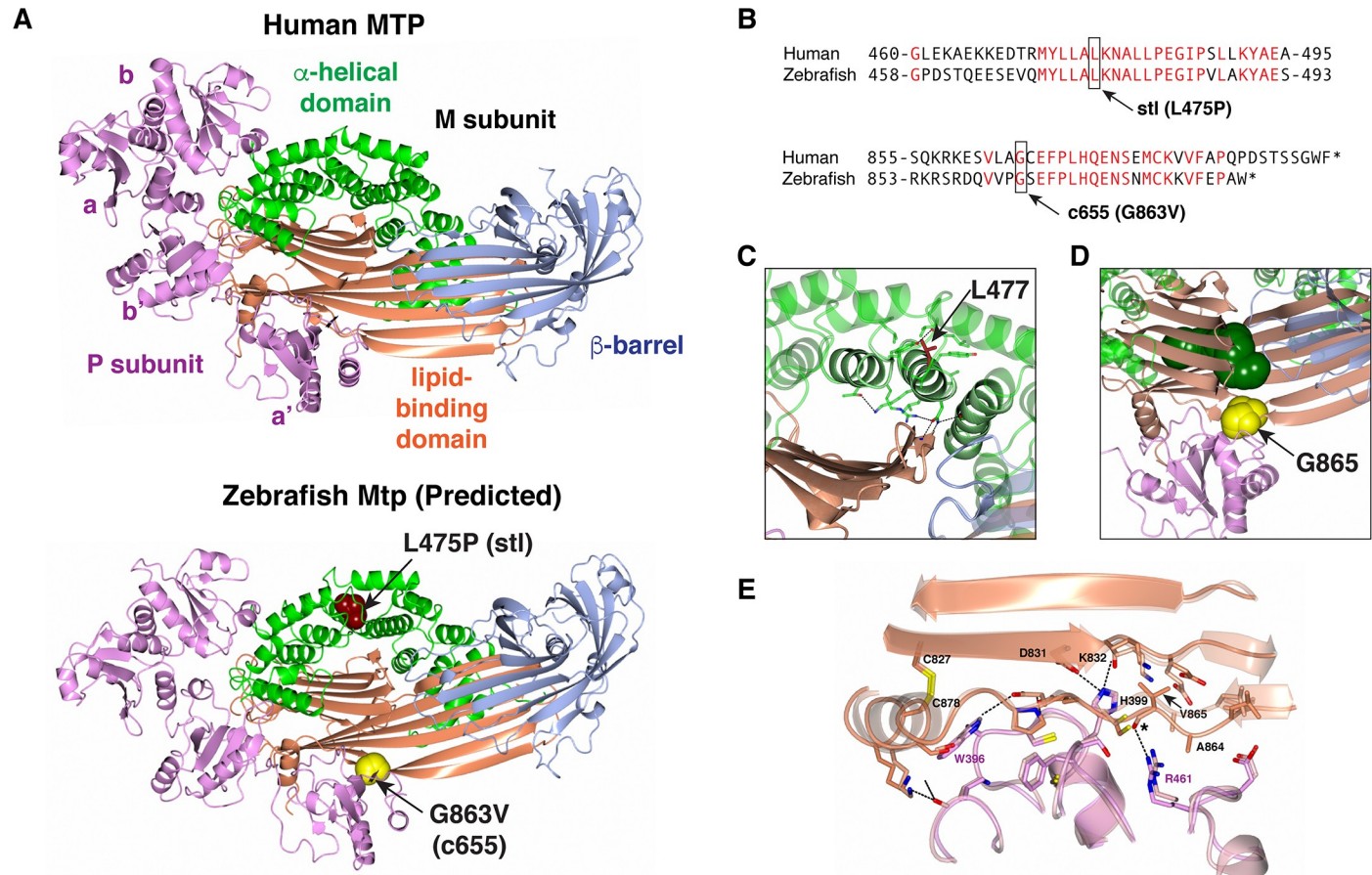

**Fig 7. Structural analysis of MTP mutations.** (A) Ribbon representation of the human MTP complex (PDB entry 6I7S) and the Zebrafish modeled structure. The positions of L475 and G863 in the Zebrafish structure are shown in space-filling representation. (B) Alignment of human MTP and zebrafish Mtp amino acid sequences surrounding the *stl* and *c655* mutations. (C) Close-up view of the area surrounding L477 in the human MTP complex. The position of L477 (red) is highlighted. The conserved hydrogen bonds linking the helical domain to the tip of the C-sheet of the lipid-binding domain are shown as well as amino acids within 4Å of L477. (D) Close-up view of the area surrounding G865 in the human MTP complex. The position of G865 (yellow) and the PEG molecule (dark green) which occupies the lipid-binding site in the solved structure are shown in space-filling representation. The **a'** domain of PDI (pink) in the complex occludes the lipid entry/exit site. (E) Close-up view showing the outer strand displacement in sheet A of the lipid-binding domain of the M subunit resulting from the G865V mutation. Asterisk indicates the wild-type backbone carbonyl of G865 hydrogen bonded to R461 of PDI. Panels C–E are colored as in panel A.

PDI **a'** domain (Fig 7D). Modeling of the zebrafish G863V (and equivalent human G865V) missense mutations suggests that the acquisition of the valine residue does not result in any gross changes in tertiary or quaternary structure of MTP. Rather, it results in a minor local displacement of the outer β-strand of the A-sheet (maximal displacement of Cα residues of 1.5 Å). It also alters the interaction between the two subunits (Fig 7E, S14 Fig).

## Discussion

The zebrafish *mttp* G863V mutation provides the first evidence that the TG and PL transfer functions of Mtp can be decoupled. The residual PL transfer activity of the mutant protein is sufficient to support secretion of small B-lps, which prevents the intestinal fat malabsorption and growth defects found when both TG and PL transfer activities of Mtp are attenuated.

It was entirely unexpected that TG and PL transfer activity in the vertebrate protein could be decoupled. Our previous analysis of MTP orthologues from divergent species, including nematodes, insects, fish, and mammals, indicated that all orthologues form a complex with PDI, localize to the ER, and support human APOB secretion [64]. However, only vertebrate MTP orthologues exhibit TG transfer activity, suggesting that PL transfer activity was the original function of MTP orthologues and that neutral lipid transfer first evolved in fish [64]. Modeling and sequence comparisons of invertebrate and vertebrate orthologues of MTTP strongly suggested that acquisition of TG transfer activity during evolution was the result of many changes in the lipid-binding cavity [64], so it was unexpected that one missense mutation in the C-terminus selectively eliminated TG transfer activity. Moreover, all of the characterized missense mutations from patients with abetalipoproteinemia that express full-length protein have been shown to be deficient in both PL and TG transfer activities [30, 62, 63, 65–73].

However, the phenotype we observed when only PL transfer was present is consistent with our previous work showing that the PL-rich high-density B-lps produced by the *Drosophila* Mttp (which only has PL transfer activity) in hepatocytes partially restore plasma lipid levels and reduce liver steatosis in a liver-specific *Mttp*-null mouse model [55]. Here, we show that transfer of PL and production of small B-lps in the *c655* mutant fish is not only sufficient for moving lipid from the liver, but is also capable of moving sufficient dietary lipid and fat-soluble vitamins from the intestine to prevent intestinal steatosis and support normal growth (Fig 4). The retention of PL transfer may also be important independent of lipoprotein production. For example, MTP-dependent PL transfer has been shown to be important for biogenesis and cell surface expression of CD1d and possibly other lipid-antigen-presenting molecules [74].

The production of B-lps in the ER of the intestine and liver is thought to occur in two steps. In the first step, MTP transfers lipids to APOB as it is translated to form small primordial particles. In the second step, it has been suggested that fusion of APOB-free lipid droplets in the lumen of the ER expands the lipoprotein core ("core expansion") [75–78]. There is evidence to suggest that MTP is also responsible for producing these ER-lumenal lipid droplets [79]. Using our LipoGlo assays, we have shown that the *c655* mutant fish produce small, homogenous particles, whereas the wild-type embryos form VLDL-sized lipoproteins in the YSL at 2–3 dpf (Fig 3D). We have made similar observations in liver-specific *Mttp* KO mice expressing *Drosophila* Mttp, which has robust PL transfer activity, but is deficient in TG transfer [55]. Expression of *Drosophila* Mttp resulted only in production of small B-lps, but human MTTP rescued the particle size [55]. Therefore, the PL transfer activity of MTP may be crucial in the generation of the small homogenous particles representative of the first step of lipoprotein assembly, whereas TG transfer might be primarily responsible for core expansion.

It is intriguing to consider that the phenotypic differences in lipid droplet sizes within the YSL between the MTP mutant alleles may be due in part to the differing availability of PL. In the *c655* mutants, it is conceivable that there is less PL available, potentially favoring larger lipid droplets that store more neutral lipid relative to the surface of the phospholipid coat. This hypothesis is consistent with data from a forward genetic screen in *Drosophila* S2 cells for modifiers of lipid droplet size and number wherein Guo *et al.* found that loss of genes critical for PL synthesis produced large lipid droplets [80]. In the current study, we have measured lipids only in adult organs; in future studies, we hope to develop methods to measure YSL lipid levels and directly test this hypothesis.

The discovery that TG and PL transfer activities can be dissociated argues for the existence of two distinct lipid binding sites in the MTP complex. Consistent with previous studies [81], the lipid-binding site identified in the human MTP crystal structure is a β-sandwich formed by the A- and C- beta-sheets in the C-terminal domain of the M subunit [16]. No charged patches were found on the inner surface of this lipid-binding site that would accommodate charged head groups of PLs, suggesting that this site is responsible primarily for binding and transfer of neutral lipid [16]. Although biochemical evidence from bovine MTP suggests that there is a second site that primarily binds PL [23], this putative site could not be identified in the current structure of human MTP [16].

Based on the crystal structure, it appears that the β-sandwich in the C-terminal domain needs to "open" in order to release and accept lipids. While it is not clear how this opening occurs, one hypothesis is that the **a'** domain of PDI moves allowing access to the lipid-binding site from the side. The G865 residue lies at the interface between the **a'** domain of PDI and residues in the lipid-binding domain of the M subunit, including strands β5 and β6 of the A-sheet, β1 of the C-sheet, and the C-terminal alpha-helix. The G865V mutation would alter the M subunit-PDI interaction, which we hypothesize may prevent movement of the **a'** domain and block access to the lipid-binding site, thereby attenuating TG binding and transfer (and perhaps transfer of other neutral lipids).

The selective loss of TG transfer activity resulting from the G865V mutation is consistent with the hypothesis that the β-sandwich solely within the C-terminal domain predominantly transfers neutral lipids. This supports the argument that a second, distinct, binding site for PL must be present. Although this additional binding site is not clear from the crystal structure, the G865V mutation does not eliminate the activity of this site, suggesting that it may be located in a different region of the protein. A putative second (PL) lipid-binding site was previously noted in the structure of the related protein, lamprey lipovitellin, at the junction between the N and A β-sheets [82]. Future work will be needed to determine whether PL and other charged lipids may bind in an equivalent location between the N-terminal half β-barrel and the A-sheet of the C-terminal domain in MTP. In this study, we used PE as a substrate for the PL transfer assays. While prior work established that MTP PL transfer is not affected by the specific PL head group [21], we have not formally ruled out the possibility that the mutation confers some specificity for different phospholipid species. In future studies, we also want to further explore the G865V substrate transfer specificities beyond the two lipids used in this study (TG & PE).

As noted above, none of the missense mutations identified in patients with abetalipoproteinemia have been found to dissociate the lipid transfer activities of MTP (reviewed in [30]). However, given that adult *c655* mutant zebrafish are indistinguishable from wild-type siblings, it is entirely possible that humans carrying a missense mutation that results in loss of TG and retention of PL transfer exist in the population. Since one copy of wild-type MTP is sufficient to prevent lipid malabsorption when faced with an oral lipid load [83], individuals heterozygous for a mutation similar to c655 may not present with any pathological changes in plasma

lipid profiles. Our data in zebrafish also suggests that individuals homozygous for a similar mutation to c655 would likely have low plasma TGs, but be otherwise normal. A search of publicly available large human GWAS databases (Global BioBank Engine, T2D Knowledge Portal, GTEx Portal) did not reveal any coding variants near G865 other than the G865X mutation known to prevent binding to PDI and loss of all MTP activity [66].

Abnormally elevated levels of APOB-containing lipoproteins and remnants promote atherosclerosis, the leading cause of death in the United States [84]. Inhibition of MTP has long been considered a possible therapeutic target for lowering disease risk by inhibiting the production of VLDL and chylomicrons [85, 86] (for review see [30, 87]). Currently, the only MTP inhibitor approved for use in patients is lomitapide (Juxtapid®, BMS-201038), which is a derivative of a compound discovered in a high-throughput chemical library screen for MTP inhibitors [85, 86]. Although its binding site is not known, lomitapide inhibits both TG and PL transfer and reduces APOB secretion [88] (Fig 6E and 6F). This drug effectively reduces LDL cholesterol, total cholesterol, and plasma APOB levels, but it is only approved for patients with homozygous familial hypercholesterolemia, whose plasma cholesterol and TG levels are up to four times the normal levels resulting in premature cardiovascular disease [89–91]. While lomitapide effectively lowers circulating lipid levels and reduces cardiovascular disease risk in these patients, side effects include fat accumulation in the liver and adverse gastrointestinal events [91–93].

The lack of intestinal and hepatic steatosis in the *c655* mutant fish suggests that an MTP inhibitor that selectively targets TG transfer activity could potentially lower plasma lipids while preventing gastrointestinal and liver side effects. This would not only improve the quality of life for patients currently taking lomitapide, but may also expand MTP inhibitor use to patients other than those with familial hypercholesterolemia. While one of the originally discovered MTP inhibitors, BMS-200150, was very effective at inhibiting TG transfer but less effective (~30%) at inhibiting PL transfer *in vitro* [86], later studies on purified MTP protein indicated the compound inhibits transfer of both lipid classes [94] and that it was not effective in animal models [85]. Now that we appreciate that the TG and PL transfer functions of MTP can be dissociated, we argue that it may be worth re-evaluating the PL transfer activity of any previously identified compounds that inhibited TG transfer activity of MTP, but failed to inhibit APOB secretion *in vitro*. Additionally, new compounds may now be designed to specifically block the TG binding site and/or alter the interaction of M subunit and the **a'** domain of PDI to mimic the G865V mutation.

In conclusion, the unexpected discovery of the *c655* missense mutation in zebrafish *mttp* has provided novel insight into the structure-function relationship of MTP, underlining the importance of forward-genetic screening approaches to reveal aspects of biology that may otherwise be missed. Our work provides the first evidence that the TG and PL transfer functions of MTP can be decoupled and that selective retention of PL transfer is sufficient for dietary fat absorption and normal growth. These results argue that selective pharmacological inhibition of TG transfer by MTP might be an effective strategy to treat hyperlipidemia.

## Methods & materials

### Ethics statement

All procedures using zebrafish were approved by the Carnegie Institution Department of Embryology Animal Care and Use Committee (Protocol #139).

### Zebrafish husbandry and maintenance

Adult zebrafish (*Danio rerio*) were maintained at 27˚C on a 14:10 h light:dark cycle and fed once daily with ~3.5% body weight Gemma Micro 500 (Skretting USA). Embryos were

obtained by natural spawning and were raised in embryo medium at 28.5°C and kept on a 14:10 h light:dark cycle. All embryos used for experiments were obtained from pair-wise crosses and were staged according to [95]. Exogenous food was provided starting at 5.5 days post fertilization (dpf) unless otherwise noted. Larvae were fed with GEMMA Micro 75 (Skretting) 3x a day until 14 dpf, GEMMA Micro 150 3x a day + Artemia 1x daily from 15 dpf–42 dpf and then GEMMA Micro 500 daily supplemented once a week with Artemia. The nutritional content of GEMMA Micro is as follows: Protein 59%; Lipids 14%; Fiber 0.2%; Ash 14%; Phosphorus 1.3%; Calcium 1.5%; Sodium 0.7%; Vitamin A 23000 IU/kg; Vitamin D3 2800 IU/ kg; Vitamin C 1000 mg/kg; Vitamin E 400 mg/kg. Zebrafish sex is not determined until the juvenile stage, so sex is not a variable in experiments with embryos and larvae. Sex of adult fish included in analyses is noted in figure legends.

*Stalactite* (*stl*) *mttp* mutant zebrafish in the *Tg(fli1:eGFP)*[y1] background [49, 96, 97] were provided by Karina Yaniv (Weizmann Institute of Science, Israel) and out-crossed to the AB wild-type strain. The *stl* mutation was maintained in both the presence and absence of the *fli1*: *eGFP* transgene. The *c655* phenotype was identified in the Farber laboratory in the background of a *kif7* mutant strain that was obtained from Philip Ingham (Lee Kong Chian School of Medicine, Singapore). The *c655 mttp* mutation was isolated from the *kif7* mutation by out-crossing to the AB wild-type strain. The *c655* mutation was crossed into the *Tg(fli1:eGFP)*[Y1] reporter line. Both *stl* and *c655 mttp* mutations were crossed into the *ApoBb.1-NanoLuc* LipoGlo reporter line [48].

## Positional cloning

To map the location of the mutation responsible for the c655 phenotype, 23 embryos with normal yolks and 23 embryos with opaque yolks (3 dpf) were processed for RNA-seq [98]. RNA was extracted from embryos by mechanical lysis in RLT buffer (Qiagen, 79216) containing 1 μL of 14.3 M beta-mercaptoethanol (Sigma, M6250). The lysate was combined with 1.8 volumes of Agencourt RNAClean XP (Beckman Coulter, A63987) beads and allowed to bind for 10 min. The plate was applied to a plate magnet (Invitrogen) until the solution cleared and the supernatant was removed without disturbing the beads. This was followed by washing the beads three times with 70% ethanol. After the last wash, the pellet was allowed to air dry for 10 min and then resuspended in 50 μl of RNAse-free water. RNA was eluted from the beads by applying the plate to the magnetic rack. RNA was quantified using the Quant-iT 610 RNA assay (Invitrogen, Q33140). Total RNA from individual embryos was DNase treated for 20 min at 37°C followed by addition of 1 μL 0.5M EDTA and inactivation at 75°C for 10 min to remove residual DNA. RNA was then cleaned using 2 volumes of Agencourt RNAClean XP (Beckman Coulter, A63987) beads under the standard protocol. Strand-specific RNA-seq libraries containing unique index sequences in the adapter were generated simultaneously following the dUTP method using 700 ng total RNA and ERCC spike mix 2 (Ambion, 4456740). Libraries were pooled and sequenced on Illumina HiSeq 2500 in 75 bp paired-end mode. Sequence data were deposited in European Nucleotide Archive under accession ERP023267. FASTQ files were aligned to the GRCz10 reference genome using TopHat2 [99] (v2.0.13, options:—library-type fr-firststrand). Ensembl 88 gene models were supplied to TopHat2 to aid transcriptome mapping. MMAPPR [51] was used to determine the location of the causal mutation. Variants were called from the pooled data using the GATK HaplotypeCaller [100]. Variants inside the regions output by MMAPPR were selected and filtered for ones where the mutant sample was called as being homozygous alternate and the siblings were heterozygous. The consequences of these variants on annotated genes was calculated using the Ensembl Variant Effect Predictor [52] and SIFT [53]. Variants with the following consequences were

selected as candidates for the causal mutation: stop_gained, splice_donor_variant, splice_acceptor_variant, transcript_ablation, frameshift_variant, stop_lost, initiator_codon_variant, missense_variant, inframe_insertion, inframe_deletion, transcript_amplification, splice_region_variant, incomplete_terminal_codon_variant.

## DNA extraction and genotyping

Genomic DNA was extracted from embryos or adult fin clips using a modified version of the HotSHOT DNA extraction protocol [101]. Embryos/tissues were heated to 95˚C for 18 min in 100 μL of 50 mM NaOH. The solution was cooled to 25˚C and neutralized with 10 μL of 1 M Tris-HCl pH 8.0. Genotyping primers for the *stalactite* allele were designed using the dCAPS Finder 2.0 program [102] and synthesized by Eurofins Genomics. The *stalactite* locus was amplified using the forward primer 5'-GTC TGA GGT TCA GAT GTA CCT GTT AGG AC-3' and reverse primer 5'-CTC TGC TGT GAT GAG CGC AGG-3' (0.5 μM primer, $T_a$ = 60˚C, extension time 30 s). The forward primer introduces an AvaII restriction site into the mutant amplicon, such that following digestion (5 units of AvaII (New England BioLabs, R0153) at 37˚C, 4 h) the WT band is 157 bp, homozygous mutants have bands at 129 bp and 28 bp, and heterozygotes have all three bands. The *c655* locus was amplified using the forward primer 5'-AGAGACGGTGTCCAAGCAGG-3' and reverse primer 5'-GCTCAAAGACTTTCTTGC-3' (0.25 μM primer, Ta = 50˚C, extension time 30 s). The *c655* mutation introduces a BsrI restriction site into the amplicon, such that following digestion (3 units of BsrI (New England Bio-Labs, R0527) in New England BioLabs Buffer 3.1 (B7203), 65˚C, 3.5 h) the WT band is 137 bp, homozygous mutants have bands at 76 bp and 61 bp, and heterozygotes have all three bands. For the *ApoBb.1-NanoLuc* genotyping protocol, see [48].

## Rescue of *c655* opaque yolk phenotype

*mttp*$^{c655/c655}$ embryos were injected at the 1-cell stage with 20 pg of zebrafish *mttp*-FLAG plasmid and 20 pg of CMV:*eGFP-CAAX* (synthesized using the Tol2kit Gateway cloning system using the p5E-CMV/SP6, pME-*eGFP-CAAX*, and p3E-polyA entry clones [103]) as a marker of successful injections. Embryos were raised to 3 dpf and screened for eGFP expression in the yolk sac. Images of eGFP+ control and experimental embryos were blinded and scored for yolk opacity by another member of the lab.

## Ectopic sprout analysis

*mttp*$^{stl/stl}$, *mttp*$^{c655/c655}$ and WT zebrafish in the *Tg(fli1:eGFP)*$^{y1}$ background were imaged at 3 dpf with a Zeiss Axiozoom V16 microscope equipped with a Zeiss PlanNeoFluar Z 1x/0.25 FWD 56 mm objective, AxioCam MRm camera, and Zen 2.5 software. The length of ectopic angiogenic segments that extend from the subintestinal vessels were analyzed in Fiji [104] (ImageJ V2.0.0, National Institutes of Health (NIH), USA) as described by [49].

## Transmission electron microscopy

Wild-type, *mttp*$^{stl/stl}$, *mttp*$^{c655/c655}$, and *mttp*$^{stl/c655}$ mutant zebrafish embryos were fixed at 4 dpf in a 3% glutaraldehyde, 1% formaldehyde, 0.1 M cacodylate solution for 1–3 h. Embryos were trimmed and swim bladders were deflated before embedding in 2% low melt agarose and processed as described in [105]. Post-fixation was performed for 1 h with 1% osmium tetroxide + 1.25% potassium ferricyanide in cacodylate solution. Following 2 x 10 min washes with water, samples were incubated with 0.05 M maleate pH 6.5 for 10 min. Samples were stained en bloc with 0.5% uranyl acetate in maleate for 4˚C overnight. Following 2 x 15 min washes

with water, samples were dehydrated through graded EtOH dilution (35%, 2 x 15 min; 50%, 15 min; 75%, 15 min; 95%, 15 min; 100% 4 x 15 min). Samples were washed with propylene oxide 4 x 15 min before incubation with 1:1 propylene oxide/resin (Epon 812 epoxy, Ladd Research Industries, Williston, VT) for 1 h and evaporated overnight. This was followed by 2 x 1 h washes in 100% resin and a final embedding in 100% resin at 55°C overnight followed by 70°C for three days. Sections were made on a Reichert Ultracut-S (Leica Microsystems), mounted on naked 200 thin mesh grids, and stained with lead citrate. Images were obtained with a Phillips Technai-12 electron microscope (FEI, Hillsboro, OR) and 794 Gatan multiscan CCD camera (Gatan, Pleasanton, CA) using Digital Micrograph software. Lipid droplet number and area was quantified with Fiji.

## Growth time-course

Unsorted embryos from pair-wise in-crosses of *stalactite* or *c655* heterozygous fish and pair-wise crosses of $mttp^{stl/+}$ x $mttp^{c655/+}$ were raised and were analyzed for standard length at 1, 3, 6, 9, 12, and 24 weeks post fertilization. At one week, fish were imaged using a Nikon SMZ1500 microscope with HR Plan Apo 1x WD 54 objective, Infinity 3 Lumenera camera and Infinity Analyze 6.5 software. Standard length [106] was measured using Fiji. Starting at three weeks, standard length was measured with a ruler. Mass of the fish was also measured starting at 6 weeks. At one and three weeks, gDNA was obtained from whole fish for genotyping. At later time-points, genotyping was performed on fin clips. Images of fish at 12 weeks post fertilization were taken with a Canon T6 camera with a Canon EF 100mm Macro Lens.

## Tissue histology

Adult zebrafish (7.5 mo; 2 males, 1 female per genotype) were placed individually into mating tanks and fasted overnight (~24 h). Fish were euthanized by submersion in ice-water. A piece of the anterior intestine and the liver were dissected from each animal and fixed in neutral-buffered formalin (Sigma, F8775) at 4°C for 48 h. Sectioning and hematoxylin & eosin staining was performed by the Johns Hopkins University Oncology Tissue Services. Slides were imaged with a Nikon E800 microscope with 60×/1.4 oil Plan Apo Nikon objective and Canon EOS T3 camera using EOS Utility image acquisition software.

## LipoGlo assays

All LipoGlo assays were performed with fish carrying a single copy of the LipoGlo ($apoBb.1^{N-luc/+}$) reporter. For detailed LipoGlo methods see [48]; Nano-Glo reporter system reagents are all from Promega Corp., (N1110; [107]). For quantitative assays and B-lp size analysis, individual embryos were dispensed into 96-well plates (USA Scientific, #1402–8589) and homogenized in 100 μL of B-lp stabilization buffer (40 mM EGTA, pH 8.0, 20% sucrose + cOmplete mini, EDTA-free protease inhibitor (Sigma, 11836170001)) by sonication with a microplate-horn sonicator (Qsonica Q700 sonicator with a Misonix CL-334 microplate horn assembly). Homogenate was stored on ice for immediate use or frozen at -20°C for later use. ApoBb.1-NanoLuc levels were quantified by mixing 40 μL of embryo homogenate with an equal volume of diluted NanoLuc buffer (1:3 Nano-Glo buffer:PBS + 0.5% NanoLuc substrate (furimazine)) in a 96-well opaque white OptiPlate (Perkin-Elmer, 6005290), and the plate was read within 2 min of buffer addition using a SpectraMax M5 plate reader (Molecular Devices) set to top-read chemiluminescent detection with a 500 ms integration time. To quantify the size distribution of B-lps, 12 μL of homogenate was combined with 3 μL of 5x loading dye (40% sucrose, 0.25% bromophenol blue, in Tris/Borate/EDTA (TBE) buffer), and 12.5 μL of the resulting solution (10% larval homogenate) was loaded per well on a 3% native polyacrylamide gel. Each

gel included a migration standard of Di-I-labeled human LDL (L3482, Thermo Fisher Scientific). Gels were run at 50 V for 30 min, followed by 125 V for 2 h. Following application of 1 mL of TBE supplemented with 2 μL of Nano-Glo substrate to the surface of the gel and incubation for 5 min, gels were imaged with an Odyssey Fc (LI-COR Biosciences) gel imaging system. Images were obtained in the chemiluminescent channel (2 min exposure) and then the 600 nm channel (30 s) for NanoLuc detection and Di-I LDL standard detection, respectively. Each lane on the gel was converted to a plot profile in Fiji and divided into LDL, IDL, VLDL and Zero Mobility (ZM) bins based on migration relative to the Di-I LDL standard. Pixel intensity from the plot profile was summed within each bin for comparison between genotypes. To determine the localization of B-lps in the whole fish, intact embryos or larvae were anesthetized and fixed in 4% paraformaldehyde for 3 h at room temperature. Following rinses in PBS + 0.1% tween-20 (3 x 15 min), embryos were mounted in 1% low-melt agarose (BP160-100, Fisher Scientific) in TBE supplemented with 1% Nano-Glo substrate. Chemiluminescent images (10 and 30 s exposures with no illumination) and a brightfield image were taken with a Zeiss Axiozoom V16 microscope equipped with a Zeiss Plan NeoFluar Z 1x/0.25 FWD 56 mm objective, AxioCam MRm camera, and Zen 2.5 software, using 2x2 binning and 2x gain. Images were quantified using Fiji; regions of interest (ROI) were drawn on the brightfield image (viscera, trunk, and head), and these ROIs were used to quantify the NanoLuc intensity on the 30 s exposure chemiluminescent images. ROIs of the same shape were used to calculate the background signal, which was subtracted from the intensity value for each ROI.

## ApoBb.1-Nluc western blotting

Protein extraction was performed on 10 pooled 3 dpf larvae per sample. Larvae were homogenized in 100 μL of 1x RIPA buffer (Millipore Sigma, 20–188) containing 3× protease inhibitor cocktail (Thermo Fisher Scientific, A32955) using a pellet pestle, and incubated at 4°C for 15 min with shaking. Samples were then centrifuged at $12,000 \times g$ for 5 min and the supernatant was mixed with an equal volume of 2× Laemmli buffer (Bio-Rad, 1610737) and heated to 95°C for 5 min. DiI-LDL (L3482, Thermo Fisher Scientific) was diluted 100-fold in RIPA buffer and extracted as above to be used as an indicator of the migration pattern of APOB, and Halo-Tagged NanoLuc protein (Promega, custom synthesized, CS188401, ~54.2kDa) was diluted 10,000-fold in RIPA buffer and used as an indicator of the migration of free NanoLuc protein. Precision Plus Protein All Blue Prestained Protein Standards (Bio-Rad, 1610373) was used as a molecular weight marker.

Twenty-five microliters of the resulting sample was loaded onto a precast 4–20% gradient gel (Bio-Rad, 4561093) and separated at 70 V for 30 min and 90 V for 60 min. Proteins were then transferred to a PVDF membrane with the Trans-blot Turbo Transfer System (Bio-Rad, 1704150) using a custom transfer program optimized to ensure transfer of high-molecular weight proteins (1.3 A constant for 15 min). The blot was blocked in 5% milk for 1 h, and then probed simultaneously with primary antibodies binding NanoLuc (R&D Systems, MAB10026-100, 1:200 dilution) and human APOB (Meridian Life Sciences, K45253G, 1:400 dilution) overnight at 4°C in 2.5% milk. The blot was then rinsed four times for 5 min each in TBST, and probed with fluorescent secondary antibodies (LI-COR Biosciences, IRDye 800CW Donkey Anti-Goat IgG, 925–32214, and IRDye 680RD Donkey Anti-Mouse IgG, 925–68072, 1:5000 dilution) for 2 h at room temperature in 2.5% milk. The blot was then rinsed as above and imaged in the 700 and 800 nm channels for 2 min each using the Odyssey Fc (LI-COR Biosciences).

## Oil Red O staining

Zebrafish larvae at 15 dpf were fixed with 4% paraformaldehyde in PBS for 3 h at room temperature and then overnight at 4°C. Fish were rinsed in 60% 2-propanol for 10 min, rocking

and then put into 0.3% Oil Red O (Sigma-Aldrich, #O0625) in 60% 2-propanol to rock overnight at room temperature. Fish were rinsed 3 times with 60% 2-propanol for 15 min. Washed fish were equilibrated step-wise into glycerol and imaged with incident light using a Nikon SMZ1500 microscope with HR Plan Apo 1x WD 54 objective, Infinity 3 Lumenera camera, and Infinity Analyze 6.5 software.

## Tissue lipid extractions, LC-MS/MS lipidomics

Adult zebrafish (1 yr.; 3 males per genotype) were fasted overnight (~24 h) and euthanized by submersion in ice-water. Similar size pieces of the anterior intestine and the liver were dissected from each animal and frozen on dry ice. Tissues were sonicated in 500 μL of homogenization buffer (20 mM Tris-HCl, 1 mM EDTA), and the protein concentration of each sample was measured using the BCA protein assay kit (Pierce, 23225). Lipids were extracted from the remaining sample volume based on equal protein concentration by a modified Bligh-Dyer procedure [108] and dried under nitrogen. Samples were analyzed by mass spectrometry at the Harvard Center for Mass Spectrometry (massspec.fas.harvard.edu). Samples were resuspended (100 μl chloroform) and analyzed on a Thermo Scientific q-Exactive Plus mass spectrometer coupled to an Ultimate 3000 uHPLC. The mass spectrometer was operated in positive ion mode for the detection of TGs and other lipids using data dependent MS/MS of top 5 peaks based on relative abundance, and using dynamic exclusion. A Dikma BioBond C4 column (50 mm, 4.6 mm, 5 μm particle size) was used with an injection volume of 15 μL. Mobile phase A was 5 mM ammonium formate, 0.1% formic acid, 5% methanol, in water v/v, and mobile phase B was 5 mM ammonium formate, 0.1% formic acid, 5% water, 35% methanol, 60% isopropyl alcohol. The column was maintained at room temperature, and gradient elution was performed as follows: 0% B for the first 5 min at 0.1 mL/min, then increased over 0.1 min to 20% B and a linear gradient to 100% B applied for 49.9 min at a flow rate of 0.4 mL/min. The solvent composition was maintained at 100% B for the next 7.9 min and then returned to 0% B over 0.1 min and maintained for the following 9 min for re-equilibration of the column prior to the next injection, all at a flow rate of 0.5 mL/min. Each MS/MS peak was curated and integrated automatically using LipidSearch 4.1 (ThermoFisher). The integration was visually inspected and integration was performed manually when needed. The 1,050 lipid species quantitated belonged to 27 classes and were further grouped into 9 categories. Heatmaps were generated using pheatmap v1.0.12 [109], after calculating the mean area of the three replicates and then taking the $\log_2$ of (mutant + 1) / (WT + 1). Lipid species that changed significantly were detected using R 3.6.1 [110] using Welch's unequal variances t-test and controlling the false discovery rate at 0.2 using the Benjamini-Hochberg procedure [111].

## Generation of *mttp*-FLAG and APOB48 plasmids

The wild-type zebrafish *mttp* coding sequence with a FLAG-tag prior to the termination codon at the C-terminus was generated by custom gene synthesis and cloned into the pcDNA3.1+ vector (*mttp*-FLAG) (Gene Universal Inc., Newark, DE). The *stl* and *c655* mutations were subsequently introduced to this plasmid by site-directed mutagenesis (Gene Universal Inc.) to generate *mttp^stl*-FLAG and *mttp^c655*-FLAG plasmids. The human pcDNA3.1-*MTTP*-FLAG plasmid was synthesized as described previously [94, 112]. The human equivalent of the *c655* mutation (G865V) was introduced into this plasmid using the Q5 Site-directed mutagenesis kit (New England Biolabs, E0554S), with the following primer pair: Forward 5'-CGTATTAGCAgtaTGTGAATTCC-3', Reverse 5- CTTTCTTTTCTTTTCTGAGAG-3'. The human APOB48 sequence [113] was cloned into the pcDNA3 under control of the CMV promoter.

## APOB secretion assays

Monkey kidney COS-7 cells (which do not express MTTP or APOB) were plated in 10 cm$^2$ cell culture dishes at a density of 9 x 10$^5$ cells per plate and grown in Dulbecco's modified Eagle's medium (DMEM) containing 10% fetal bovine serum, L-glutamine, and antibiotics at 37˚C. COS-7 cells were transfected with 5 μg of plasmid expressing human APOB48 cDNA under the control of CMV promoter using EndoFectin (Genecopoeia, EF014) according to the manufacturer's protocol. After 24 hours, cells from each dish were harvested, equally distributed in 6-well plates, and reverse transfected with 3 μg of either pcDNA3, pcDNA3-*mttp*-FLAG, pcDNA3-*mttp$^{stl}$*-FLAG, pcDNA3-*mttp$^{c655}$*-FLAG, pcDNA3-*MTTP*-FLAG, or pcDNA3-*MTTP* (G865V)-FLAG plasmids. After 32 h cells were incubated overnight with 1 mL of DMEM containing 10% FBS. The overnight conditioned media were collected to measure APOB by ELISA [113, 114]. Cells were scraped in PBS and a small aliquot was used to measure total protein using a Coomassie protein assay (Thermo Scientific, #1856209). Cells were lysed in cell extract buffer (100 mM Tris, pH 7.4, 150 mM NaCl, 1 mM EGTA, 1mM EDTA, 1% Triton X-100, 0.5% sodium deoxycholate). Lysates were rotated for 1 h at 4˚C to solubilize the membranes and centrifuged at 16,000g for 30 min. APOB was measured in the supernatant via ELISA. Briefly, high binding 96-well plates (Corning, #3366) were incubated with capture antibody anti-LDL (APOB), clone 1D1 (MyBiosource, #MBS465020, 1:1000 dilution) overnight at room temperature. The plate was washed 3x with PBS-T (PBS + 0.05% Tween-20), blocked with 3% BSA (Boston Bio Products, #P753) for 1 h and washed 3x with PBS-T, before incubating with 100 μL of standards and experimental samples for 3 h. The plate was washed 3x with PBST and incubated with 100 μL of human APOB antibody (Academy Bio-Medical Company, Inc., #20S-G2, 1:1000 dilution) for 1 h. After washing the plate 3x with PBS-T, 100 μL of alkaline phosphatase labeled anti-goat IgG (Southern Biotech, #6300–04, 1:3000 dilution) was added to each well and incubated for 1 h. The plate was washed 3x with Diethanolamine buffer, pH 9.5 and 100 μL of PNPP (Thermo Scientific, 34045, 1 mg/mL) was added to each well before reading the plate at 405 nm in a PerkinElmer Victor$^3$ 1420 multilabel counter. APOB concentrations in the media and in cells was normalized to μg total cell protein for each sample. Data for zebrafish and human plasmids were obtained in the same experiments, but are graphed separately in Figs 5A, 5B & 6C, 6D; the pcDNA3 control data is displayed in both sets of graphs.

## Immunofluorescence

COS-7 cells were plated at a density of 50,000 cells on coverslips in 12-well dishes and transfected with 2 μg of plasmids expressing either zebrafish or human MTTP-FLAG plasmids. After 48 h, cells were fixed in paraformaldehyde and blocked with PBS supplemented with 1 mM MgCl$_2$, 0.5 mM CaCl$_2$, 3% BSA, 0.1% Triton X-100, and 1% horse serum. Cells were incubated with anti-FLAG M2 monoclonal antibody (Sigma # F3165, 1:250 dilution) and anti-calnexin antibody (Santa Cruz Biotechnology, # sc-11397, 1:250 dilution) overnight. Cells were washed three times with PBS and incubated with goat anti-mouse Alexa Fluor-594 (Invitrogen, #A11005, 1:500 dilution) and donkey anti-rabbit Alexa-Fluor-488 (Invitrogen, # A21206, 1:500 dilution) for 1 h. The cells were washed and mounted with Vectashield mounting medium (Vector Laboratories, #H-1000). Images were taken on a Leica SP5II confocal microscope with a 63x1.4 HCX PL Apo oil immersion lens.

## Immunoprecipitation and western blotting

Transfected COS-7 cells were washed three times with ice cold PBS and scraped in buffer K (1 mM Tris-HCl, 1 mM EGTA and 1 mM MgCl$_2$, pH 7.6) containing protease inhibitor cocktail

(Sigma, # P2714). Cells were mechanically lysed by passing them 10 times with $30_{1/2}$-gauge needle and small fractions were used to measure the total protein using a Coomassie protein assay (Thermo Fisher Scientific, #1856209). Cell lysate was incubated with Anti-FLAG M2 antibody for 1 h and immunoprecipitated (IP) using (protein A/G) agarose beads (Santa-Cruz Biotechnology, # SC2003). The supernatants were used to detect actin via western blotting and served as loading controls. Both the supernatant and immunoprecipitated fractions were subjected to electrophoresis on an 8% SDS-PAGE gel. The weight separated proteins were transferred to nitrocellulose membranes and probed with either anti-FLAG M2 (1:1000) or anti-PDI (Cell Signaling Technology, #3501 (1:1000)), anti-actin (Thermo Fisher Scientific, #PA1-183, (1:3000)) prepared in 2% BSA in TBS. The blots were washed and probed with HRP-conjugated corresponding secondary antibodies (goat anti-rabbit, Cell Signaling Technology, #7074, 1:5000 or goat anti-mouse, Thermo Fisher Scientific, #62–6520, 1:5000). The blots were developed in ChemiDoc[TM]-Touch Imaging system from Bio Rad.

### Triglyceride transfer assay

Following transfection with plasmids as described above, cell lysate (35 μg) prepared in buffer K containing protease inhibitor cocktail was incubated with donor vesicles containing NBD-labeled triolein (Setareh Biotech, LLC, #6285) and acceptor vesicles. Fluorescence was measured at different time intervals (5, 10, 15, 30, 45 and 60 min). Percent TG transfer was calculated after subtracting the blank and dividing it by the total fluorescence reading obtained by disrupting vesicles with isopropanol, as described previously [19, 20]. Where noted, assays also included the MTTP inhibitor lomitapide (Aegerion Pharmaceuticals, #AEGR-733) at a concentration of 1 μM.

### Phospholipid transfer assay

COS-7 cells were transfected with 9 μg of either zebrafish *mttp*-FLAG or human *MTTP*-FLAG plasmids in 10 cm² cell culture dishes. After 48 h, cell lysates were prepared in buffer K containing protease inhibitor cocktail (Sigma, #P2714). The cell lysates were centrifuged at 12,000g for 10 min at 4˚C. A small aliquot of cell lysate was used for measuring protein and kept for western blotting to measure expression level. Equal concentrations of protein from each sample (400 μg) were incubated with 40 μL of M2 agarose beads (Sigma, #A2220) for 3 h at 4˚C. FLAG-tagged proteins were eluted in 150 μL of buffer K containing 2 μL of 150 ng/μL FLAG peptide (Sigma, #F3290; 1 h at 4˚C). PL transfer activity was assayed using nitrobenzoxadiazole (NBD)-labeled Phosphoethanolamine, triethylammonium (Thermo Fisher Scientific, #N360). The purified FLAG-tagged proteins (100 μL) were incubated with donor vesicles containing NBD-Phosphoethanolamine and acceptor vesicles. The fluorescence was measured at different time intervals (0, 1, 2, 3, and 4 h). The percentage transfer of PL was calculated as the difference between the fluorescence reading at the 0 h time point and 3 h time point divided by the total fluorescence reading obtained by disrupting vesicles with isopropanol as described previously [19, 20]. Where noted, assays also included the MTTP inhibitor Lomitapide at a concentration of 1 μM.

### Modeling

Predicted models of zebrafish Mttp and zebrafish PDI were generated based on the human MTP complex (PDB entry 6I7S) [16] using *SWISS-MODELLER* [115]. The zebrafish Mtp complex was prepared by superposition of the zebrafish models of Mttp and PDI to the coordinates of the human MTP complex with LSQ superpose tool of the graphics program *Coot* [116] and posterior energetical minimization with the geometry minimization program of the *Phenix*

*Suite* [117]. Mutations of human and zebrafish residues G865V/G863V and L477P/L475P were generated manually with the graphics program *Coot*, the resulting complexes being energetically minimized as described for the model of zebrafish MTP. The figures were generated with *CCP4mg* [118].

## Statistical analyses

Graphing and some statistics, including One-way, randomized block and Repeated Measures ANOVA with Bonferroni post-hoc tests, Kruskall-Wallis with Dunn's Multiple Comparison test and Chi-square tests were performed with GraphPad Prism (GraphPad Software). When sample sizes and variance between groups were significantly different, Robust ANOVA was performed using R to determine overall significance of noted datasets [119](https://cran.r-project.org/web/packages/WRS2/vignettes/WRS2.pdf), [120], (https://rcompanion.org/rcompanion/d_08a.html). When significant differences were present between genotypes, Games-Howell post-hoc tests were used to make pair-wise comparisons at each time point using SPSS Statistics (IBM), adjusting the significance level for multiple comparisons. Details of the statistical analyses can be found either in the figure legend or results sections. Sample sizes for each experiment are indicated in the figure legends for each experiment.

## Additional software

DNA, mRNA, and protein sequence alignments were performed with MacVector V15.5 (MacVector, Inc.). Microsoft Word and Excel were used for manuscript preparation and data analysis, respectively, figures were assembled in Adobe Illustrator CS5 (Adobe Systems) and references were assembled with EndNote 8X.

## Supporting information

**S1 File. Single nucleotide variants present in *c655* mutant embryos.**
(XLSX)

**S2 File. Lipidomics raw data.**
(XLSX)

**S3 File. Significantly different lipids between genotypes.**
(XLSX)

**S4 File. Source Data.**
(XLSX)

**S1 Fig. *c655* mutant yolks appear dark with transmitted light and off-white with incident light.** (A) Wild-type and *mttp^c655/c655* mutant embryos were imaged at 3 dpf using either transmitted light (illumination below the fish) or incident light (illumination from above the fish). The wild-type embryos are translucent; the pigment cells on the opposite side of the embryo (red arrow) are visible through the yolk with both light sources. The yolk is opaque in the mutants; it appears dark with transmitted light and off-white with incident light. Pigment cells on the opposite side of the embryo are barely visible in mutant embryos, regardless of light source. Scale = 200 μM.
(TIF)

**S2 Fig. Expression of wild-type zebrafish *mttp*-FLAG rescues the opaque yolk phenotype in *mttp^c655/c655* embryos.** One-cell stage *mttp^c655/c655* embryos were co-injected with CMV: *mttp*-FLAG and the CMV: *eGFP-CAAX* plasmid, or CMV: *eGFP-CAAX* alone as a control. Embryos

expressing eGFP-CAAX in the YSL were imaged at 3 dpf, and images were scored for the degree of yolk opacity by a lab member who was blinded to the genotype of the fish. (A) Representative image of an $mttp^{c655/c655}$ mutant embryo expressing eGFP-CAAX in the YSL and a fully opaque yolk. (B) Examples of injected embryos with varying degrees of yolk opacity (normal translucent yolk, opaque region in the yolk extension, opaque patches in the anterior yolk with or without opaque yolk extension). (C) Images were binned into the four noted categories of yolk opacity. Results represent pooled data from 3 independent experiments, n = 91 control and 102 Mtp-FLAG eGFP-positive embryos total. Chi-square test, $p < 0.001$. Scale = 500 µM. (TIF)

**S3 Fig. The *stl* and *c655 mttp* mutations have differential effects on the degree of yolk opacity during embryonic development.** Representative images of wild-type, $mttp^{stl/stl}$, $mttp^{c655/c655}$ and trans-heterozygous $mttp^{stl/c655}$ mutants from 1 dpf to 6 dpf. The $mttp^{stl/stl}$ mutants are visibly opaque at 1 dpf and the area of opacity is retained for longer than in $mttp^{stl/c655}$ or $mttp^{c655/c655}$ mutants. Images at 3 dpf are the same fish shown in Fig 1. Scale = 500 µM. (TIF)

**S4 Fig. Lipid droplets block light transmission through the larval intestine.** (A) Wild-type fish at 6 dpf were fed a high-fat meal for 1 h, as described previously [57]. Unfed fish have translucent intestines (black arrow, left) when imaged with transmitted light, whereas fed fish have opaque intestines (black arrow, right). Scale = 500 µM. (B) Electron microscopy following a 1 h high-fat feed reveals an accumulation of cytoplasmic lipid droplets in the intestinal enterocytes. By scattering light and blocking light transmission through the intestine, the accumulation of cytoplasmic lipid droplets causes the intestine to appear opaque. Nucleus (n), mitochondria (mito), brush border (bb), lipid droplet (LD). Scale = 10 µM. (TIF)

**S5 Fig. ApoBb.1-Nluc is full length in *mttp* mutants.** Representative immunoblot for the NanoLuc reporter in wild-type and *mttp* mutant zebrafish embryos. The NanoLuc reporter is fused to the C-terminus of the zebrafish *apoBb.1* gene. Lanes represent lysate from 10 pooled 3 dpf *mttp* wild-type, $mttp^{c655/c655}$, and $mttp^{stl/stl}$ mutant embryos, as well as wild-type AB embryos that do not carry the NanoLuc reporter. Purified Halo-tagged NanoLuc protein (Nluc Halo-tag, ~54 kDa) was used as a positive control for NanoLuc and DiI-LDL was used to mark the migration of APOB. Blot was probed simultaneously for NanoLuc (magenta) and Human APOB (green). The ApoBb.1-NanoLuc is exclusively detected as a high molecular weight band (>250 kDa) corresponding to the migration of human APOB. Note that the APOB antibody does not recognize zebrafish ApoB. (TIF)

**S6 Fig. LipoGlo lipoprotein gel primary data.** Original gels corresponding to the data in Fig 3D. Each gel shows a composite image of the fluorescent DiI-LDL migration standard (yellow) and LipoGlo emission chemiluminescent exposure (blue) from WT, $mttp^{stl/stl}$, and $mttp^{c655/c655}$ fish. Gels were analyzed as detailed in [48] and lipoprotein particles were binned into four classes based on migration relative to the DiI-LDL standard, including zero mobility (ZM), and three classes of serum B-lps (VLDL, IDL and LDL). (TIF)

**S7 Fig. *p*-values associated with Fig 3D.** (TIF)

**S8 Fig. Developmental time-course of standard length and mass measurements of *mttp* mutant fish and siblings.** Results are representative of pooled data from two independent

experiments, n = 7–80 fish/genotype/time-point, mean +/- SD. Significance was determined with a Robust ANOVA and Games-Howell post-hoc tests were used to make pair-wise comparisons at each time point. Using a Bonferroni correction, p-values were adjusted to control for multiple comparisons (6 length or 4 mass comparisons), a: *stl/+* vs. *stl/stl*, $p < 0.01$, b: *+/+* vs. *stl/stl* and *stl/+* vs. *stl/stl*, $p < 0.05$, c: *+/+* vs. *stl/stl* and *stl/+* vs. *stl/stl*, $p < 0.001$, d: *stl/+* vs. *stl/stl*, $p < 0.05$, e: *+/+* vs. *stl/stl*, $p < 0.01$, f: *stl/+* vs. *stl/stl*, $p < 0.001$.
(TIF)

**S9 Fig. *c655* mutant embryos exhibit fewer ectopic angiogenic segments extending from the subintestinal vessels than *stl* mutant embryos.** (A) The developing vasculature is visualized in the *Tg(fli:eGFP)$^{y1}$* transgenic zebrafish line [97]. The subintestinal vessels (boxed region) grow bilaterally onto the dorsolateral surface of the yolk sac. Scale = 200 μM. (B) Representative wide-field images of *Tg(fli:eGFP)$^{y1}$* in wild-type, *mttp$^{stl/stl}$*, or *mttp$^{c655/c655}$* embryos at 3.5 dpf. Ectopic sprouts extending ventrally from the subintestinal vein are more common in *mttp$^{stl/stl}$* mutant embryos than in *mttp$^{c655/c655}$* mutant embryos. Scale = 200 μM. (C) Quantification of the average number of ectopic sprouts in *mttp* mutants and siblings on 3.5 dpf. Results represent pooled data from 3 independent experiments, n = 28–36 total embryos/genotype group; mean +/- SD, Kruskall-Wallis with Dunn's Multiple Comparison test, \*\*\* $p < 0.001$.
(TIF)

**S10 Fig. Significant lipid accumulation in the intestine of *stl* but not in *c655* mutants.** Representative images of isolated intestines from adult WT and *mttp* mutant fish (7.5 mo), scale = 1 mm.
(TIF)

**S11 Fig. Additional LC-MS lipidomics data.** (A) Tissue lipid extracts from WT and *mttp* mutant lines were quantitated by LC-MS/MS and grouped into lipid classes and expressed as a sum of lipid group (n = 3). (B) The number of individual lipid species in the different lipid classes that are statistically different from WT in the intestine (I) or liver (L) (adjusted *p*-value < 0.20). Triacylglycerol (TG), diacylglycerol (DG), monoacylglycerol (MG), phosphatidylcholine (PC), phosphatidylethanolamine (PE), phosphatidylglycerol (PG), phosphatidylinositol (PI), methylphosphocholine (MePC), lysophosphatidylserine (LPS, lysophosphatidylethanolamine (LPE), lysophosphatidylcholine (LPC), bismethyl phosphatidic acid (BisMePA), sphingomyelin (SM), cholesterol ester (CE), ceramides (Cer), monoglycosylceramide (CerG1), acyl carnitine (AcCa).
(TIF)

**S12 Fig. Triglyceride and phospholipid transfer assay time-course data with zebrafish proteins.** (A, B) Measurements for TG (A) and PL transfer (B) by zebrafish Mtp-FLAG and mutant proteins over a time-course. The single time-points depicted in the bar graphs of Fig 5E & 5F, correspond to the 45 min and 180 min (TG and PL transfer, respectively) time-points in the curves shown. For both, n = 3 (each n is the mean of three technical replicates from independent experiments), mean +/- SD, Repeated Measures ANOVA with Bonferroni post-hoc tests, significance as noted in figure. (C) Representative western blot of immunoprecipitated and eluted Mtp-FLAG proteins from COS-7 cell lysate used in the PL transfer assays. COS-7 cells transfected with FLAG-tagged proteins were immunoprecipitated from cell lysates using anti-FLAG antibodies and eluted with FLAG peptides. Blot on eluted fractions indicates equal concentrations of the various Mtp-FLAG proteins; actin blot indicates equal loading of cell lysate.
(TIF)

**S13 Fig. Related to Fig 6.** COS-7 cells were transfected and FLAG-tagged human MTTP proteins were immunoprecipitated from cell lysates using anti-FLAG antibodies and eluted with FLAG peptides. Representative western blot on eluted fractions indicates equal concentrations of the various MTP-FLAG proteins; actin blot indicates equal loading of cell lysate.
(TIF)

**S14 Fig. Views of the interaction between the lipid-binding domain of the M subunit and the a' domain of PDI.** Views show the same region as in Fig 7E. (A) Wild-type human, (B) mutant V865 human, and (C) wild-type zebrafish shown separately. (D) Overlay of wild-type human and wild-type zebrafish. In (A), the backbone carbonyl of G865 is shown hydrogen-bonded to R461 of PDI (arrow).
(TIF)

## Acknowledgments

We gratefully acknowledge Michael Sepanski for electron microscopy, Andrew Rock, Carmen Tull, Julia Baer, and Mackenzie Klemek for fish husbandry, Matthew Bray for assistance using R, and Amy Kowalski for synthesis of the pDESTTol2pA2-CMV: *eGFP-CAAX* plasmid, as well as Jennifer Anderson and Tabea Moll for help editing the manuscript. We would also like to especially thank Philip Ingham, who provided the *kif7* mutant strain in which we identified the *c655* mutation, and Sunia A. Trauger, of the Harvard Center for Mass Spectrometry for performing the lipidomics analysis.

## Author Contributions

**Conceptualization:** Meredith H. Wilson, M. Mahmood Hussain, Steven A. Farber.

**Formal analysis:** Meredith H. Wilson, Sujith Rajan, Aidan Danoff, Richard J. White, Monica R. Hensley, Vanessa H. Quinlivan, Rosario Recacha, James H. Thierer, Frederick J. Tan.

**Funding acquisition:** Meredith H. Wilson, Sujith Rajan, James H. Thierer, Elisabeth M. Busch-Nentwich, Lloyd Ruddock, M. Mahmood Hussain, Steven A. Farber.

**Investigation:** Meredith H. Wilson, Sujith Rajan, Aidan Danoff, Richard J. White, Monica R. Hensley, Vanessa H. Quinlivan, James H. Thierer.

**Methodology:** Meredith H. Wilson, James H. Thierer, M. Mahmood Hussain.

**Project administration:** Meredith H. Wilson, M. Mahmood Hussain, Steven A. Farber.

**Resources:** James H. Thierer, Elisabeth M. Busch-Nentwich, Lloyd Ruddock, M. Mahmood Hussain, Steven A. Farber.

**Software:** Richard J. White, Lloyd Ruddock.

**Supervision:** Elisabeth M. Busch-Nentwich, Lloyd Ruddock, M. Mahmood Hussain, Steven A. Farber.

**Validation:** Meredith H. Wilson, M. Mahmood Hussain, Steven A. Farber.

**Visualization:** Meredith H. Wilson, Sujith Rajan, Richard J. White, Rosario Recacha, Frederick J. Tan, Lloyd Ruddock.

**Writing – original draft:** Meredith H. Wilson, Sujith Rajan, Richard J. White, Vanessa H. Quinlivan, James H. Thierer, Elisabeth M. Busch-Nentwich, Lloyd Ruddock, M. Mahmood Hussain, Steven A. Farber.

**Writing – review & editing:** Meredith H. Wilson, Sujith Rajan, Richard J. White, Monica R. Hensley, Vanessa H. Quinlivan, James H. Thierer, Elisabeth M. Busch-Nentwich, Lloyd Ruddock, M. Mahmood Hussain, Steven A. Farber.

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
