## [Decision Letter · Decision Letter 0]

22 Feb 2020

Dear Dr Farber,

Thank you very much for submitting your Research Article entitled 'A point mutation decouples the lipid transfer activities of microsomal triglyceride transfer protein' to PLOS Genetics. Your manuscript was fully evaluated at the editorial level and by independent peer reviewers. The reviewers appreciated the attention to an important problem, but raised some substantial concerns about the current manuscript. Based on the reviews, we will not be able to accept this version of the manuscript, but we would be willing to review again a much-revised version. We cannot, of course, promise publication at that time.

If you decide to revise the manuscript for further consideration at PLOS Genetics, please aim to resubmit within the next 60 days, unless it will take extra time to address the concerns of the reviewers, in which case we would appreciate an expected resubmission date by email to plosgenetics@plos.org.

[LINK]

We are sorry that we cannot be more positive about your manuscript at this stage. Please do not hesitate to contact us if you have any concerns or questions.

Yours sincerely,

Daniel Rader

Guest Editor

PLOS Genetics

Gregory Barsh

Editor-in-Chief

PLOS Genetics

Reviewer's Responses to Questions

**Comments to the Authors:**

Reviewer #1: 1. Does the mutation (zebrafish G863V or human G865V) alter the transfer of any other lipid besides triglyceride, e.g., cholesterol ester, sphingolipid, diacylglycerol?

2. p. 11, lines 293-296 – the authors write “this finding supports the biochemical evidence for an additional “slow” phospholipid transfer site (42) and provides additional evidence that this site is distinct from the lipid-binding cavity formed by the…..” The data are consistent with an additional phospholipid transfer site, but the authors do not present any data concerning the relative rate of transfer at that site. I would recommend removing the word “slow”.

3. Fig. 3 E, F – the authors do not comment on the results of the studies using the MTTP inhibitor lomitapide. It is interesting that lomitapide inhibits phospholipid transfer to the same degree in both the wild-type and mutant proteins suggesting it blocks transfer at both sites (if there are two). Some MTP inhibitors work presumably by blocking the lipid binding cavity in the C-terminal domain. Have you tried any other MTP inhibitors to see if you get the same result as you did with lomitapide? If you used an inhibitor that blocks solely at the lipid binding cavity, you might expect that the PL transfer activity in the G865V variant would not be affected. This doesn’t impact the conclusion that the phospholipid and triglyceride transfer can be decoupled; however, it does raise questions about the mechanism(s) of phospholipid transfer.

4. Fig. 4 – the results of the experiments with the LipoGlo PAGE Gels suggest that the mttpc655/c655 embryos produce few VLDL particles; however, they do produce a significant number of particles in the LDL range. These particles must have core neutral lipid. If triglyceride transfer is blocked, what is in the core lipid in these particles? Cholesteryl ester? (This goes back to the question raised in point #1 above.)

5. Fig. 5 – There is a large difference in size of lipid droplets in the YSL of the mttpc655/c655 embryos compared to those in either the mttpstl/stl or the trans-heterozygotes. The authors speculate that retained neutral lipid is stored in larger lipid droplets in the YSL of mttpc655/c655 embryos because phospholipid is being secreted and consequently is less available to coat lipid droplets. Some basic arithmetic casts doubt on that hypothesis. A single droplet with a radius of 5 microns has the same surface area (phospholipid) as 25 droplets that have a radius of 1 micron. Consequently, it’s a bit of a stretch to say there is less phospholipid available in the mttpc655/c655 embryos without some quantitative data, which is beyond the scope of this investigation. The observation, nonetheless, is quite interesting and could be very important.

Minor comments:

• p. 2 – line 46 – “vassals” should be vessels.

• p. 2 – line 48 – why not say “humans” instead of “people”?

• p. 2, line 51 – why do you use the term “fats” for lipids? Is this to be a summary for the general public without scientific terminology? If it is not, why not say triglyceride?

• p. 3, line 81 – “ApoB” should be “APOB” by your convention.

• p.3, line 105 – there should be the word “on” between “based the” i.e., “based on the”

• p. 4, line 110 – “primary” should be “primarily”

• p. 18, line 473 – the word “present” should be “presents”

Reviewer #2: Please take my criticism constructively. There are good aspects to the study, which in this reviewers mind need to be pulled out to reach a wider audience.

Abstract-

The authors are assuming that the readership of PLOS genetics will know about Mttp-and there is no reason why they should. Really not a good Abstract to sell your story-only people with specialist knowledge/interest would be encouraged to read this manuscript. You are under-selling your work. What is the unique selling point of your study design? What is it a good example of?

Introduction

I feel that the authors haven’t provided the Introduction in terms of why they did what they did. In contrast to most papers (which put the Introduction in the Discussion), you have put the Discussion in the Introduction, perhaps? The Introduction does not set you up for what you have done. Why, did you start from the point you have? What was your motivation? Almost the first sentence of the Results is the first sentence of the Introduction? Why were you searching zebrafish for defects in yolk utilization-why was this important to do? You talk a great deal about PDI in the results and yet this is not mentioned.

Results

1). Lack of apoB secretion-phraseology correct but for readers of Plos Genetics and the phenotype you are seeing would come across better if you link it up with a defect in lipid secretion, what is the evidence here that the lipid is only being secreted on ApoB-lipoproteins-again given the conclusion you are drawing-perhaps the Introduction should remind us what the chief lipids in the yolk sac of fish are? And how they are exported out of the yolk. Do fish have VTG?

2). Have you checked the lipid composition of the egg-yolk in the mutants? This is sort of addressed in Figure 5 but even so.

3). Bottom of Page 7/top of 8, expecting to see the result before new section heading-perhaps, they need to remove to under the next section heading.

4). I would liked to have been told what Lomitapide was developed to inhibit and given the structure of MTTP where it is envisaged to bind.

5). I think with a better Introduction, you could present your results in a more logical order. What is the evidence for the statement ‘During this time, the fish are relying solely on yolk lipids’ Also what is the evidence for the next statement.

6). The phospholipid transfer activity based on just one phospholipid species-is it the most suitable one. How do you know the MTTP:PDI complex is binding triglyceride?

A lot of data but doesn’t quite grab this reviewers attention-because a clear story is not coming through. Just wonder whether the problem started from the Introduction-I feel you are not really getting across why the study is important-you present some of the key results at the beginning and then things rather fade away. The novelty of your apoB assay could come earlier to grab our attention, perhaps. I think if you reduced the description of your results by at least 20%, you would get the main points across more effectively. I would encourage you have more regular subheadings: in some cases they could say more.

**Have all data underlying the figures and results presented in the manuscript been provided?**

Reviewer #1: Yes

Reviewer #2: Yes

PLOS authors have the option to publish the peer review history of their article (what does this mean?). If published, this will include your full peer review and any attached files.

Reviewer #1: No

Reviewer #2: No

---

## [Decision Letter · Decision Letter 1]

10 Jun 2020

Dear Dr Farber,

Thank you very much for submitting your Research Article entitled 'A point mutation decouples the lipid transfer activities of microsomal triglyceride transfer protein' to PLOS Genetics. Your manuscript was fully evaluated at the editorial level and by independent peer reviewers. The manuscript will very likely be acceptable but the editors would like you to address the remaining minor issues raised by Reviewer #2.

We therefore ask you to modify the manuscript according to the review recommendations before we accept your manuscript. Your revisions should address the specific points made by Reviewer #2.  Extensive revision or re-writing is not necessary, rather please attempt to respond to the specific points made.

[LINK]

Yours sincerely,

Daniel Rader

Guest Editor

PLOS Genetics

Gregory Barsh

Editor-in-Chief

PLOS Genetics

Reviewer's Responses to Questions

**Comments to the Authors:**

Reviewer #1: Thank you for the thoughtful responses to my initial comments.

Reviewer #2: Abstract

1) Much Improved

Results:

1). Could you please tell the reader the nature of the other MTTP variant found (predicted to be tolerated in Table S1)

2). Figure 4, Second paragraph in Section c665 mutant adults are largely protected from Intestinal steatosis is still a problem. I became rather bored here (all rather woolly) until I looked at the Figure and then realised the key aspects of the data are not spelt out for the reader. I feel the author spent too much time justifying what they did (First two sentences) rather than getting to the point and focusing in on the lipidomic analysis, which is key to linking into the next section. The data appear equivocal (potentially due to small sample sizes) but I think the authors data in the liver on the TG and DG in their c655 mice are consistent with their hypothesis. What is not different is as potentially important result as was is potentially different. Would be good to corroborate TG/DG data on the yolk.

3). On page 14, you say you were surprised to find that the c655-surely that was what you were expecting from the lipidomic analysis & lipoprotein analysis? Myself, I would be reassured rather than surprised by this result-corroborating the arguably weak lipidomic data. You are not quite making the link with the lipidomic data, the results in this section are almost backwards. The PDI aspect is interesting but perhaps best after you tie up the lipid aspects. Then in would be easier to link up the PDI story with the structural data. Manuscript still too turgid in places for this impatient reviewer.

4). I think you send too much time on Fig 6- could almost be Supplementary Figure-just say it recapitulates the zebrafish data and point out significance of inhibitor.

I think Discussion could be shorter-and rejigging the presentation of your data (From Fig 4 onwards) will facilitate this.

**Have all data underlying the figures and results presented in the manuscript been provided?**

Reviewer #1: Yes

Reviewer #2: Yes

PLOS authors have the option to publish the peer review history of their article (what does this mean?). If published, this will include your full peer review and any attached files.

Reviewer #1: No

Reviewer #2: No

---

## [Editor Report · Decision Letter 2]

17 Jun 2020

Dear Dr Farber,

We are pleased to inform you that your manuscript entitled "A point mutation decouples the lipid transfer activities of microsomal triglyceride transfer protein" has been editorially accepted for publication in PLOS Genetics. Congratulations!

Yours sincerely,

Daniel Rader

Guest Editor

PLOS Genetics

Gregory Barsh

Editor-in-Chief

PLOS Genetics

Comments from the reviewers (if applicable):

**Data Deposition**

http://datadryad.org/submit?journalID=pgenetics&manu=PGENETICS-D-20-00021R2

**Press Queries**

---

## [Editor Report · Acceptance letter]

28 Jul 2020

PGENETICS-D-20-00021R2 

A point mutation decouples the lipid transfer activities of microsomal triglyceride transfer protein 

Dear Dr Farber, 

We are pleased to inform you that your manuscript entitled "A point mutation decouples the lipid transfer activities of microsomal triglyceride transfer protein" has been formally accepted for publication in PLOS Genetics! Your manuscript is now with our production department and you will be notified of the publication date in due course.

With kind regards,

Kaitlin Butler

PLOS Genetics

On behalf of:
